# Taxonomic Reappraisal of Periconiaceae with the Description of Three New *Periconia* Species from China

**DOI:** 10.3390/jof8030243

**Published:** 2022-02-28

**Authors:** Er-Fu Yang, Rungtiwa Phookamsak, Hong-Bo Jiang, Saowaluck Tibpromma, Darbhe J. Bhat, Samantha C. Karunarathna, Dong-Qin Dai, Jian-Chu Xu, Itthayakorn Promputtha

**Affiliations:** 1Center for Yunnan Plateau Biological Resources Protection and Utilization, College of Biological Resource and Food Engineering, Qujing Normal University, Qujing 655011, China; erfu20170431@gmail.com (E.-F.Y.); saowaluckfai@gmail.com (S.T.); samanthakarunarathna@gmail.com (S.C.K.); 2Department of Biology, Faculty of Science, Chiang Mai University, Chiang Mai 50200, Thailand; 3Research Center in Bioresources for Agriculture, Industry and Medicine, Chiang Mai University, Chiang Mai 50200, Thailand; 4Honghe Center for Mountain Futures, Kunming Institute of Botany, Chinese Academy of Sciences, Honghe 654400, China; rphookamsak@outlook.com (R.P.); hongbo-J@hotmail.com (H.-B.J.); jxu@mail.kib.ac.cn (J.-C.X.); 5Master of Science Program in Applied Microbiology (International Program), Faculty of Science, Chiang Mai University, Chiang Mai 50200, Thailand; 6CIFOR-ICRAF China Program, World Agroforestry (ICRAF), Kunming 650201, China; 7Centre for Mountain Futures (CMF), Kunming Institute of Botany, Kunming 650201, China; 8Center of Excellence in Fungal Research, Mae Fah Luang University, Chiang Rai 57100, Thailand; 9School of Science, Mae Fah Luang University, Chiang Rai 57100, Thailand; 10No. 128/1-J, Azad Housing Society, Curca, P.O. Box, Goa Velha 403108, India; bhatdj@gmail.com

**Keywords:** Ascomycota, *Bambusistroma*, Chinese mycobiota, Dothideomycetes, *Noosia*

## Abstract

As a result of an ongoing research survey of microfungi in Yunnan, China, several saprobic ascomycetes were collected from various host substrates. Preliminary morphological analyses identified a few of these taxa as *Periconia* species. We obtained DNA sequence data of the *Periconia* species from pure cultures and investigated their phylogenetic affinities. Phylogenetic analyses of a combined LSU, ITS, SSU and *tef1-α* sequence dataset demonstrated that five isolates of *Periconia* formed well-resolved subclades within Periconiaceae. Accordingly, three new *Periconia* species are introduced viz. *P. artemisiae*, *P. chimonanthi* and *P. thysanolaenae*, and new host and geographical records of *P. byssoides* and *P. pseudobyssoides*, are also reported from dead branches of *Prunus armeniaca* and *Scrophularia ningpoensis*. *Periconia celtidis* formed a monophyletic clade with *P. byssoides* in the present phylogenetic analyses. Results of the pairwise homoplasy index (PHI) test indicated significant recombination between *P. byssoides* and *P. celtidis*. Therefore, *P. celtidis* has been synonymized under *P. byssoides*. In addition, we re-illustrated and studied the type specimen of the sexual genus *Bambusistroma*. As a type species of *Bambusistroma*, *B. didymosporum* features similar morphology to the sexual morph of *Periconia homothallica* and *P. pseudodigitata*. We therefore synonymize *Bambusistroma* under *Periconia* based on morphological and phylogenetic evidence. Furthermore, our new isolates produced brown conidia of asexual morph in agar media typical of the genus *Noosia*. Based on morphological comparison with *Periconia* in vitro and phylogenetic status of *Noosia*, we also treat *Noosia* as a synonym of *Periconia*. Detailed descriptions and illustrations of three novel taxa and two new records of *Periconia byssoides* and *P. pseudobyssoides* as well as the illustration of *P. didymosporum* comb. nov. are provided. An updated phylogenetic tree of Periconiaceae using maximum likelihood and Bayesian inference analyses is constructed. Generic circumscription of *Periconia* is amended.

## 1. Introduction

Periconiaceae Nann. was introduced by Nannizzi [1] and is typified by *Periconia* Tode. The family is classified as the suborder Massarineae, order Pleosporales, class Dothideomycetes in Ascomycota [2]. Taxa in this family are well known as saprobes occurring on various hosts worldwide [2,3]. Only a few species, such as *Flavomyces fueloephazae* D.G. Knapp et al., *Periconia circinata* (L. Mangin) Sacc. and D. Sacc., *P. digitata* (Cooke) Sacc. and *P. macrospinosa* Lefebvre and Aar. G. Johnson, were reported as endophytes and pathogens [2,4,5,6]. Some *Periconia* species such as *P. atropurpurea* (Berk. and M.A. Curtis) M.A. Litv., *P. byssoides* Pers. and *P. macrospinosa* also produce valuable bioactive compounds that possess powerful pharmacological activities [2,7,8,9,10,11].

Periconiaceae is poorly studied and has long been neglected in modern taxonomic treatment [12]. Saccardo [13] initially assigned ‘Periconieae Sacc.’ to accommodate dematiaceous hyphomycetes, forming macronematous conidiophores with conidial heads and 1-celled, pigmented conidia. Later, Nannizzi [1] raised ‘Periconieae’ to the familial rank as Periconiaceae and accommodated *Periconia* and *Stachybotrys* Corda in this family. *Stachybotrys* is currently accommodated in its own family Stachybotryaceae (as ‘Stachybotriaceae’) [14]. Tanaka et al. [12] emended the family Periconiaceae (=Periconieae) and re-circumscribed the modern taxonomic description of the family in the suborder Massarineae based on molecular data coupled with morphological characteristics of the sexual-asexual morph.

Four genera, including *Bambusistroma* D.Q. Dai and K.D. Hyde [15], *Flavomyces* D.G. Knapp et al. [4], *Noosia* Crous, R.G. Shivas and McTaggart [16] and *Periconia* were accepted in the family Periconiaceae based on phylogenetic analyses [2,12,17]. However, the monotypic genera *Bambusistroma*, *Flavomyces* and *Noosia* are always cladded with *Periconia* species [18,19,20,21,22]. Tanaka et al. [12] mentioned that *Periconia* might be polyphyletic, although the generic type of *Periconia*, *P. lichenoides* Tode lacks the necessary molecular data to confirm phylogenetic affinities. Hence, the phylogenetic statuses of *Bambusistroma*, *Flavomyces* and *Noosia* require further study.

Taxa in Periconiaceae are mainly reported by their noosia-like or periconia-like asexual morphs that are characterized by macronematous, mononematous, pigmented, septate, branched, or unbranched, smooth to echinulate, thick-walled conidiophores, or sometimes lacking conidiophores; conidiophores bearing monoblastic to polyblastic, integrated or discrete conidiogenous cells on stipe and branches, with ovoid to clavate or spherical conidial heads, with small, pimple-like buds and catenated or solitary, golden brown to dark brown, globose to subglobose, or fusoid-ellipsoidal, aseptate, smooth-walled or verruculose, or echinulate conidia, sometimes with a minute, unthickened pore at the base [2,12].

The sexual morphs of Periconiaceae have only been reported for *Bambusistroma* and some species of *Periconia* such as *P. homothallica* Kaz. Tanaka and K. Hiray. and *P. pseudodigitata* Kaz. Tanaka and K. Hiray. The sexual morph of Periconiaceae is characterized by immersed to erumpent, scattered to grouped, subglobose to globose ascomata, with central, periphysate ostiolar neck, thin-walled of peridium, 8-spored, bitunicate, fissitunicate, oblong to cylindrical, short-pedicellate asci, with the well-developed ocular chamber, embedded in cellular pseudoparaphyses and hyaline, fusiform, septate, smooth-walled ascospores with an entire sheath [2,12,15].

The genus *Periconia*, introduced by Tode [23], is typified by *P. lichenoides* that was found on stems of herbaceous plants. Unfortunately, the type specimen was lost and only Tode’s drawings were left as the iconotype for morphological studies [24]. Currently, the genus has 206 epithets available in Index Fungorum (http://www.indexfungorum.org/Names/Names.asp; accessed on 20 January 2022). However, about 36 species previously placed in *Periconia* have been transferred to other genera in Dothideomycetes, Pezizomycetes, Sordariomycetes and Ascomycota genera *incertae sedis* as well as synonymized to other *Periconia* species [13,25,26,27,28,29,30,31,32,33,34,35,36,37,38,39]. To date, *Periconia* comprises 117 morphological species [39], of which only 24 species have so far been confirmed of phylogenetic affinities based on molecular data [2,40]. Species of *Periconia* are usually known as saprobes on a variety of plant substrates and have a wide distribution across temperate and tropical regions [3,24,41].

A monotypic genus *Bambusistroma* typified by *B. didymosporum* D.Q. Dai and K.D. Hyde was invalidly (art. F.5.1, Shenzhen) introduced by Adamčík et al. [15] but it was validated by Index Fungorum [42]. The genus was introduced to accommodate a sexual bambusicolous taxon from Thailand and is characterized by stromatic, uni-loculate, subglobose to conical, immersed to erumpent ascomata, with a minute papillate ostiole, 8-spored, bitunicate, cylindrical asci, and hyaline, fusiform, 1-septate ascospores, surrounded by distinct mucilaginous sheath [15]. An asexual morph of *Bambusistroma* has not yet been reported. Adamčík et al. [15] treated the genus in Massarinaceae, while Tanaka et al. [12] placed the genus in Periconiaceae based on phylogenetic evidence. Currently, only a putative species, *B. didymosporum*, is accommodated in this genus. *Bambusistroma* always formed a clade with *Flavomyces* and other *Periconia* species in Periconiaceae [12,20,21,22,43]. The phylogenetic status of *Bambusistroma*, therefore, remains questionable.

The asexual genus *Noosia* was introduced by Crous et al. [16] to accommodate the putative species *N. banksiae* Crous, R.G. Shivas and McTaggart, associated with leaf spots of *Banksia aemula* in Australia. The genus is characterized by hyaline to brown, smooth, branched mycelium, becoming verruculose and aggregated into hyphal strands with age, solitary, integrated, inconspicuous, lateral or terminal conidiogenous cells, with small, pimple-like pores, and dimorphic conidia, borne solitary or in short chains. The conidial type I is aseptate, globose, subhyaline, smooth, becoming fusoid-ellipsoidal, brown, verruculose, solitary or in short, branched chains. The conidial type II is arising as phragmoconidia from disarticulating hyphal cells, brown, verruculose apex obtuse, truncate base, with minute, unthickened pore [16]. *Noosia* often forms a clade with other *Periconia* species, and thus, the phylogenetic status of the genus also remains unclarified [12,20,21,43].

The aims of this study are to introduce three novel species, two new host and geographical records of *Periconia* based on multigene phylogenetic analyses and morphological characteristics. In addition, a taxonomic revision of two monotypic genera, *Bambusistroma* and *Noosia,* is resolved with these two genera being treated as synonyms of *Periconia* based on typical morphological characteristics and phylogenetic evidence.

## 2. Materials and Methods

### 2.1. Sample Collection, Morphological Characterization, Isolation and Preservation

Fungal specimens were collected from Chuxiong, Kunming and Xishuangbanna in Yunnan Province, China, and stored in disposable plastic Zip-loc bags. The specimens were observed and examined after 1–2 days incubation at room temperature in the laboratory. The microscopic morphological features (e.g., conidiophores, conidiogenous cells, conidia) were examined with an OLYMPUS SZ61 stereomicroscope and Nikon ECLIPSE Ni compound microscope equipped with a Canon DS126311 digital camera. In addition, *Bambusistroma didymosporum* and *Noosia banksiae* were re-illustrated. Morphological features viz. ascomata, peridium, pseudoparaphyses, asci, ascospores (sexual morph); conidiophores, conidiogenous cells and conidia (asexual morph) were measured using Tarosoft (R) Image Frame Work (IFW) version 0.9.7, and the photographic plates were processed with Adobe Photoshop CS6 Extend version 10.0 software (Adobe systems, San Jose, CA, USA).

Pure cultures of the new isolates were obtained by single spore isolation [44]. The conidial masses on the surface of fungal hosts were picked up using a sterilized surgical needle and soaked in sterilized water droplets for spore suspension. The spore suspension was spread onto the surface of potato dextrose agar plates (PDA) and incubated overnight at room temperature. Germinated conidia were singularly transferred to two new PDA plates (five conidia in each plate) and incubated at room temperature in normal day/night-light cycle. The colony developed of single conidium was transferred to new PDA plates and incubated at room temperature for one week. Culture characteristics and growth were recorded at one week and after four weeks. The in vitro sporulation was observed after three months. The pure cultures were preserved in PDA, sterilized 10% glycerol, and double-distilled water (ddH_2_O) for short- and long-term storage.

Herbarium specimens were deposited at the herbarium of Cryptogams Kunming Institute of Botany Academia Sinica (KUN-HKAS), Yunnan, China, while the pure cultures were deposited at Kunming Institute of Botany Culture Collection (KUMCC), Kunming, China. The novel taxa were registered in Index Fungorum (http://www.indexfungorum.org/names/IndexFungorumRegisterName.asp; accessed on 10 December 2021).

### 2.2. DNA Extraction and PCR Amplification

Fresh mycelia (50–100 mg) were scraped from pure colonies growing on PDA after 4 weeks and stored in 1.5 mL sterilized microcentrifuge tubes for the genomic DNA extraction. The genomic DNA was extracted using Biospin Fungus Genomic DNA Extraction Kit–BSC14S1 (BioFlux^®^, Hangzhou, China), following the protocol in the manufacturer. The genomic DNA solution was stored at 4 °C for the PCR amplification and maintained at −20 °C for long-term storage.

Genomic DNA was amplified by polymerase chain reaction (PCR) for four gene regions, including the partial 28s large subunit nuclear ribosomal DNA (LSU), the 18s small subunit nuclear ribosomal DNA (SSU), the internal transcribed spacers (ITS) and the translation elongation factor 1-alpha (*tef1-α*). The primer pairs LR0R/LR5 [45], NS1/NS4 [46], ITS5/ITS4 [46] and EF1-983F/EF1-2218R [47] were used to amplify the sequence fragments of LSU, SSU, ITS and *tef1-α*, respectively.

The reactions of PCR amplification were performed in a 25 µL total volume, containing 8.5 µL of sterilized double-distilled water (ddH_2_O), 12.5 µL of 2 × Power *Taq* PCR MasterMix (mixture of Easy*Taq*TM DNA Polymerase, dNTPs, and optimized buffer, Beijing Bio Teke Corporation (Bio Teke, Wuxi, China), 1 μL of each forward and reverse primers (10 μM) and 2 μL of DNA template (50 ng/μL). The polymerase chain reaction (PCR) thermal cycle program for ITS, LSU, SSU and *tef1-α* followed Tibpromma et al. [48]. The amplified PCR products were purified and sequenced by a commercial sequencing provider (TsingKe Biological Technology Co., Beijing, China). Newly generated sequences were deposited in GenBank (Table 1).

### 2.3. Sequence Alignment and Phylogenetic Analyses

The generated consensus sequences were obtained via BioEdit v. 7.0.9.0 [49] and subjected to a basic local alignment search tool (nucleotide BLAST; https://blast.ncbi.nlm.nih.gov/Blast.cgi; accessed on 11 June 2021) for delineating closely related taxa. Based on the nucleotide BLAST search, new isolates belong to the genus *Periconia*, and a corresponding sequence dataset of newly generated taxa and other closely related taxa in Periconiaceae (Table 1) was generated for further phylogenetic analyses based on recent publications [6,43,50,51,52].

Individual sequence datasets were aligned separately via MAFFT online version 7.475 using default settings (http://mafft.cbrc.jp/alignment/server; accessed on 25 June 2021) [53] and improved where necessary using BioEdit v. 7.0.9.0 [49]. Individual sequence datasets were prior analyzed by maximum likelihood (ML) criterion for checking the congruence of tree topologies and further phylogenetic analyses of the concatenated LSU, SSU, ITS and *tef1-α* sequence dataset were performed by maximum likelihood (ML) and Bayesian inference (BI) criteria.

Maximum likelihood (ML) was analyzed by RAxML-HPC2 on XSEDE (8.2.10) [54,55] via the CIPRES science Gateway V.3.3 web server [56] using default settings, but with adjustments by setting up the substitution evolution model as GTRGAMMAI and 1000 replicates of rapid bootstrap. The evolutionary model of nucleotide substitution was selected independently for each locus (LSU, SSU, ITS and *tef1-α*) using MrModeltest 2.3 [57]. The best-fit model is the GTR + I + G substitution model for each locus under the Akaike Information Criterion (AIC).

Bayesian inference (BI) was performed by MrBayes on XSEDE (3.2.7a) via the CIPRES science Gateway V.3.3 web server [56]. Bayesian posterior probabilities (BYPP) [58,59] were evaluated by Markov Chain Monte Carlo sampling (MCMC). Parameters were set as defaults, but with the adjustments as two parallel runs of six simultaneous Markov chains. Starting random tree topology was run with 3,000,000 generations for a combined dataset, and trees were sampled every 100 generations (resulting in 30,000 total trees). The effective sampling sizes (ESS) and the stable likelihood plateaus and burn-in value were determined by Tracer version 1.7.1 software [60]. The first 6000 sampled trees (20%) representing the burn-in were discarded and the remaining trees were used to calculate the Bayesian posterior probabilities (BYPP) in a 50% majority rule consensus tree. The final alignment and phylogram were submitted to TreeBASE (https://www.treebase.org/, accessed on 22 February 2022, submission ID: 29297). Phylogram was visualized in FigTree v. 1.4.0 (http://tree.bio.ed.ac.uk/software/figtree/; accessed on 22 February 2022) and reorganized in Microsoft Office PowerPoint 2017 (Microsoft Inc., Redmond, WA, USA).

### 2.4. Genealogical Concordance Phylogenetic Species Recognition (GCPSR) Analysis

The recombination level within phylogenetically closely related species of newly generated strains KUMCC 20-0263 and KUMCC 20-0264 with *Periconia byssoides* Pers. (MAFF 243869, MFLUCC 18-1548, MFLUCC 18-1553, MFLUCC 17-2292), *P. celtidis* Tennakoon, C.H. Kuo and K.D. Hyde (MFLUCC 20-0172, NCYUCC 19-0314), *P. pseudobyssoides* Markovsk. and A. Kačergius (DUCC 0850, MAFF 243868, MAFF 243874) and *P. salina* Dayarathne and E.B.G. Jones (MFLU 19-1235) was determined by a pairwise homoplasy index (PHI) test [61] performed in SplitsTree4 [62,63], using the GCPSR model [64]. The combined ITS, LSU, SSU and *tef1-α* sequence dataset of these phylogenetically closely related species was applied for the test to refine incompatibility between pairs of sites regarding a genealogical history does not involve recurrent or convergent mutations that can be parsimoniously inferred [61]. The split graph showing relationships between newly generated strains KUMCC 20-0263, KUMCC 20-0264 and other closely related species was visualized and constructed in LogDet transformation and splits decomposition options. The pairwise homoplasy index resulted in a threshold lower than 0.05 (Φw < 0.05), indicating significant recombination in the dataset.

## 3. Results

### 3.1. Phylogenetic Analyses

The concatenated ITS, LSU, SSU and *tef1-α* sequence dataset consisted of 69 strains of representative taxa in Periconiaceae, and the other two related families, Didymosphaeriaceae and Massarinaceae, included five new generated strains (*Periconia artemisiae* KUMCC 20-0265, *P. byssoides* KUMCC 20-0264, *P. chimonanthi* KUMCC 20-0266, *P. pseudobyssoides* KUMCC 20-0263 and *P. thysanolaenae* KUMCC 20-0262), with *Morosphaeria ramunculicola* (K.D. Hyde) Suetrong et al. (KH 220) and *M. velatispora* (K.D. Hyde and Borse) Suetrong et al. (KH 221) as the outgroup taxa. The concatenated sequence matrix comprises 4005 total characters, including gaps (651 characters for ITS, 906 characters for LSU, 1405 characters for SSU, 1043 characters for *tef1-α*). Phylogenetic investigation based on maximum likelihood analysis conducted the best RAxML tree with a final likelihood value of −17,280.650596, presented in Figure 1. The matrix had 1095 distinct alignment patterns, with 43.72% undetermined characters or gaps. Estimated base frequencies were as follows: A = 0.238246, C = 0.255371, G = 0.267517, T = 0.238865; substitution rates AC = 1.513277, AG = 2.455316, AT = 1.810928, CG = 1.226908, CT = 9.347925, GT = 1.000000; gamma distribution shape parameter α = 0.201661. The final average standard deviation of split frequencies at the end of total MCMC generations was calculated as 0.009307 in BI analysis.

Phylogenetic analyses of the concatenated ITS, LSU, SSU and *tef1-α* sequence dataset based on ML and BI were similar in overall tree topologies. *Periconia artemisiae* sp. nov. (KUMCC 20-0265) formed a distinct clade with *Periconia* sp. (strains Otu0123, G1782 and C75) with high support (100% ML, 1.00 PP; Figure 1). *Periconia artemisiae* also formed a well-resolved clade with *P. alishanica* Tennakoon, C.H. Kuo and K.D. Hyde, *P. pseudobyssoides*, *P. salina*, *P. byssoides* and *P. celtidis* with high support (100% ML, 1.00 PP). *Periconia chimonanthi* sp. nov. shared the same branch length with *Periconia* sp. CY137 (isolated from the nest of *Cyphomyrmex wheeleri* in Texas, USA) and is sister to *P. cortaderiae* Thambug. and K.D. Hyde with high support in BI analysis but shows insignificant support in ML analysis (62% ML, 0.99 PP; Figure 1). While *P. thysanolaenae* sp. nov. formed a separate branch sister to *P. minutissima* Corda strain MUT 2887 with insignificant support in ML and BI analyses (67%, 0.80 PP) and clustered with *P. homothallica* and *Noosia banksiae*.

The other two newly generated strains KUMCC 20-0263 and KUMCC 20-0264 clustered with *P. pseudobyssoides* and *P. byssoides*, respectively. The new isolate KUMCC 20-0264 grouped with *P. byssoides* (MAFF 243869, MFLUCC 17-2292, MFLUCC 18-1548, MFLUCC 18-1553), *P. celtidis* (MFLUCC 20-0172, NCYUCC 19-0314) and *P. salina* (MFLU 19-1235) with low support (Figure 1). Based on phylogenetic analyses, KUMCC 20-0264 is identified as *P. byssoides*. While the isolate KUMCC 20-0263 clustered with other isolates of *P. pseudobyssoides*, and the isolate KUMCC 20-0263 is thus identified as *P. pseudobyssoides*. Furthermore, *Bambusistroma didymosporum* formed a separate branch with *Sporidesmium tengii* W.P. Wu (HKUCC 10837), while *Noosia banksiae* is sister to *Periconia homothallica* (KT 916) with insignificant support (Figure 1).

### 3.2. Genealogical Concordance Phylogenetic Species Recognition (GCPSR)

Twelve representative strains of *Periconia byssoides*, *P. celtidis*, *P. pseudobyssoides* and *P. salina* were implied in a pairwise homoplasy index (PHI) test for determining the recombination level of these four species. The results of PHI test indicated no significant recombination among *P. byssoides*, *P. pseudobyssoides* and *P. salina* in ITS (Φw = 0.2398), LSU (Φw = 0.6534), *tef1-α* (Φw = 0.4152), and a combined ITS, LSU, SSU and *tef1-α* sequence dataset (Φw = 0.654) while SSU showed that there are too few informative characters to implemented; however, it is significant recombination between *P. byssoides* and *P. pseudobyssoides* in LSU region. PHI results of single gene ITS, LSU, *tef1-α* and a combined ITS, LSU, SSU, *tef1-α* sequence dataset indicated significant recombination between *P. byssoides* and *P. celtidis*. Therefore, these two species are shown to be conspecific and thus, *P. celtidis* is treated as a synonym of *P. byssoides* (Figure 2).

### 3.3. Taxonomy

***Periconia*** Tode, Fung. mecklenb. sel. (Lüneburg) 2: 2, emend.

Synonym:

*Bambusistroma* D.Q. Dai and K.D. Hyde, Index Fungorum 225: 1 (2015)

*Noosia* Crous, R.G. Shivas and McTaggart, in Crous, Groenewald, Shivas, Edwards, Seifert, Alfenas, Alfenas, Burgess, Carnegie, Hardy, Hiscock and Hübe, Persoonia 26: 139 (2011).

Sexual morph: Ascomata solitary to gregarious, sometimes with pseudostromatic or clypeus-like basal structure on host, scattered to clustered, immersed to erumpent, dark brown to black, globose to subglobose, or conical, ostiolate, papillate; ostiolar canal filled with hyaline periphyses. Peridium composed of several layers of thin- or thick-walled, brown to dark brown pseudoparenchymatous cells of *textura angularis* to *textura prismatica*. Hamathecium composed of numerous, cellular, hyaline, septate, branched, anastomosed pseudoparaphyses, embedded in a gelatinous matrix. Asci 8-spored, bitunicate, fissitunicate, cylindrical, with short, rounded or furcate pedicel, apically rounded with a shallow ocular chamber. Ascospores overlapping 1–2-seriate, hyaline, fusiform, 1-septate, with guttules, smooth-walled, with an entire sheath. Asexual morph: Conidiophores macronematous, mononematous, sometimes lacking, erect, straight or slightly flexuous, branched or unbranched, pale brown to dark brown, septate, smooth-walled, or slightly echinulate, sometimes swollen near the base, with spherical conidial heads on main stipe and/or apical branches. Conidiogenous cells monoblastic or polyblastic, terminal, integrated or discrete, ovoid to subglobose, or subspherical, lightly pigmented, branched, smooth to slightly echinulate, sometimes with small, pimple-like pores. Conidia solitary or catenate, in acropetal chains, globose to subglobose, aseptate, occasionally ellipsoidal to cylindrical, pale brown to dark brown, smooth-walled or verruculose. In vitro: Mycelium hyaline to brown, branched, smooth to verruculose with age, frequently aggregating into hyphal strands. Conidiophores sometimes reduced to conidiogenous cells; when present, macronematous, erect, single, sometimes in pairs, pigmented, septate, thick-walled. Conidiogenous cells mono- to polyblastic, solitary, discrete or integrated, determinate, or inconspicuous, lateral and terminal, pigmented, ellipsoidal, ovoid to clavate, aseptate, sometimes with small, pimple-like pores, or with percurrent proliferations. Conidia dimorphic; primary conidia globose to subglobose, or fusoid-ellipsoidal, subhyaline to brown or dark brown, smooth to verruculose, solitary or in short, branched chains, sometimes with minute, unthickened basal pores; secondary conidia phragmoconidia, brown, verruculose, arising from disarticulating hyphal cells (adopted from Crous et al. [16], Calvillo-Medina et al. [51]).

Type species: *Periconia lichenoides* Tode.

Notes: *Periconia* was recognized as a polyphyletic genus in Periconiaceae due to the other three monotypic genera that always clade with other *Periconia* species [12,18,19,20,21,22]. However, morphological study of *Bambusistroma* and *Noosia* demonstrated that these two genera resemble the sexual morph of *Periconia homothallica* and *P. pseudodigitata* as well as the sporulation in the culture of *P. artemisiae*, *P. chimonanthi* and *P. pseudobyssoides*. *Bambusistroma* always clustered with *Flavomyces* in previous studies [12,20,21,22,43]. In this study, we include more taxon sampling in Periconiaceae, and the phylogenetic analyses indicated that *B. didymosporum* is sister to *Sporidesmium tengii* (HKUCC 10837), clustering with other *Periconia* species with insignificant support. There is only one LSU sequence available for the non-type strain HKUCC 10837 of *S. tengii*, and therefore, the phylogenetic status of *S. tengii* remains unclarified and is pending further study. Crous et al. [16] introduced the genus *Noosia* due to the lack of conspicuous conidiophores, which is different from other *Periconia* in nature. However, we were able to induce the sporulation of *P. artemisiae*, *P. chimonanthi* and *P. pseudobyssoides* on PDA after three months. Morphologically, sporulation in the cultures of *P. artemisiae*, *P. chimonanthi* and *P. pseudobyssoides* resembles *Noosia banksiae*. Hence, in the present study, we treated *Bambusistroma* and *Noosia* as synonyms of *Periconia* based on morphological characteristics and phylogenetic analyses.

***Periconia artemisiae*** E.F. Yang, H.B. Jiang and Phookamsak, sp. nov. Figure 3.

Index Fungorum number: IF559495.

Etymology: Name reflects the host genus, *Artemisia*, from which the holotype was collected.

Holotype: KUN-HKAS 10738.

Sexual morph: Undetermined. Asexual morph: Colonies superficial, numerous, effuse, hairy, dark brown to black. Conidiophores 240–385 μm long, 13.5–18.5 μm wide, macronematous, mononematous, solitary or gregarious (1−3 together on stroma), arising from stromatic base, straight to slightly flexuous, dark brown, slightly paler towards the apex, 3−5-septate, sometimes branched at the apex, forming conidial heads at apex, spherical, up to 110 μm diam. Conidiogenous cells 11.5–13.5 × 9–11 μm ($\bar{x}$ = 12.5 × 9.7 μm, n = 15), polyblastic, brown to dark brown, spherical, obovate or subglobose, terminal, verruculose, discrete, located at nodose apices of conidiophores, with hyaline and sympodial conidiogenous loci (Figure 3f). Conidia 16–19 × 14–18 μm ($\bar{x}$ = 17.2 × 16 μm, n = 20), catenate, globose, initially faint yellow or light brown, becoming brown to dark brown at maturity, aseptate, verruculose. In vitro: Culture sporulated in PDA after three months. Mycelium immersed, consisted of 1–2 μm diam., branched, septate, smooth to slightly verruculose, hyaline to brown hyphae. Conidiophores reduced to conidiogenous cells. Conidiogenous cells 2.5–4 × 1–2 μm ($\bar{x}$ = 3 × 1.5 μm, n = 15), mono- to polyblastic, with 1–2 conidiogenous loci, lateral, or integrated, lateral and terminal, brown to dark brown, inconspicuous, giving rise to solitary conidia, or in short chains. Conidia 5–8 × 4–5 μm ($\bar{x}$ = 6.7 × 4.3 μm, n = 20), globose, brown to dark brown, aseptate, smooth to slightly verruculose.

Culture characteristics: Conidia germinated on PDA within 24 h. Colonies on PDA reaching 20 mm diam in one week at room temperature (10–25 °C), dense, circular, flattened, dull, cottony, surface smooth with entire edge, hairy at the margin, pale grey at the margin and pale brown to white at the middle from the above; greenish-brown at the margin and dark brown to black at the middle in reverse; not producing pigmentation on agar medium.

Material examined: CHINA, Yunnan Province, Kunming City, Xiehong Dam (25°8′31″ N, 102°44′15″ E, altitude 1918 msl), saprobic on dead branches of *Artemisia apiacea* Hance (Asteraceae), 11 July 2019, E.F. Yang, DY9 (KUN-HKAS 107384, holotype), ex-type living culture, KUMCC 20-0265.

Distribution: China (Yunnan).

Host and habitat: Saprobic on *Artemisia apiacea* (Asteraceae) in a terrestrial environment.

Notes: *Periconia artemisiae* matches the typical morphology of *Periconia* in forming macronematous, mononematous conidiophores, with a spherical conidial head, polyblastic conidiogenous cells, and catenate, globose, brown to dark brown, aseptate, verruculose conidia. Phylogenetic analyses based on combined ITS, LSU, SSU and *tef1-α* sequence data also supported the placement of the species in *Periconia* (Figure 1). Based on the nucleotide BLAST results of the ITS region, *P. artemisiae* is similar to the *Periconia* sp. isolate G1782 and *Periconia* sp. isolate Otu0123 with 99.80% similarity, and *Periconia* sp. isolate C75 with 99.79% similarity. *Periconia artemisiae* is also similar to *P. alishanica* (MFLUCC 19-0145, NCYU 19-0347 and NCYUCC 19-0186) with 97.65% similarity. Phylogenetic analyses based on combined ITS, LSU, SSU and *tef1-α* sequence data (Figure 1) indicated that *P. artemisiae* forms a separate branch sister to *Periconia* sp. (G1782, Otu0123, and C75) with high support and constituted basal to *P. alishanica*, *P. pseudobyssoides*, *P. salina* and *P. byssoides*. Based on a nucleotide pairwise comparison of ITS and *tef1-α* [65], *P. artemisiae* differs from *P. alishanica* strain MFLUCC 19-0145 (type strain) in 14/507 bp (2.76% of ITS) and 23/817 bp (2.82% of *tef1-α*), differs from *P. byssoides* strain MLFUCC 18-1553 in 16/501 bp (3.19% of ITS) and 24/920 bp (2.61% of *tef1-α*), differs from *P. pseudobyssoides* strain MAFF 243868 in 14/509 bp (2.75% of ITS) and 23/929 bp (2.48% of *tef1-α*), and differs from *P. salina* strain MFLU 19-1235 (type strain) in 15/509 bp (2.95% of ITS). A nucleotide base comparison of ITS regions also indicated *Periconia* sp. (G1782, Otu0123, and C75) is conspecific with *P. artemisiae* (less than 1.5% difference). Therefore, these three strains are identified as *P. artemisiae* in this study. Morphologically, *P. artemisiae* can be distinguished from *P. alishanica* by having wider and shorter conidiophores, with branched at the apex and smaller conidia [40]. *Periconia alishanica* was isolated from many host plants in Taiwan viz. *Ficus septica* (Moraceae), *Macaranga tanarius* (Euphorbiaceae) and *Morus australis* (Moraceae), whereas *P. artemisiae* was found as a saprobe on *Artemisia apiacea* (Asteraceae) in Yunnan, China. Morphological characteristics of *P. artemisiae* (Otu0123, G1782 and C75) are not available. Therefore, *P. artemisiae* (KUMCC 20-0265) and *P. artemisiae* (Otu0123, G1782 and C75) could not be compared.

***Periconia banksiae*** (Crous, R.G. Shivas and McTaggart) H.B. Jiang, Bhat and Phookamsak, comb. nov.

Index Fungorum number: IF559496.

Basionym: *Noosia banksiae* Crous, R.G. Shivas and McTaggart, in Crous, Groenewald, Shivas, Edwards, Seifert, Alfenas, Alfenas, Burgess, Carnegie, Hardy, Hiscock and Hübe, Persoonia 26: 139 (2011).

Type information: AUSTRALIA, Queensland, Noosa, S 26°34′14.0″ E 153°4′21.6″, on leaves of *Banksia aemula*, 13 July 2009, P.W. Crous, R.G. Shivas and A.R. McTaggart, CBS H-20587 (holotype), ex-type living culture CPC 17282 = CBS 129526.

Detailed description and illustration: see Crous et al. [16]

Notes: *Noosia banksiae* was introduced by Crous et al. [16] as the type species of the monotypic genus *Noosia*. The species was isolated from leaf spots of *Banksia aemula* in Australia. Crous et al. [16] introduced the genus in Pleosporales but did not classify the genus to any family in Pleosporales. According to Crous et al. [16], the nucleotide BLAST search of ITS and LSU sequence data showed that *N. banksiae* had a close relationship with *Periconia* species, *Sporidesmium tengii* (HKUCC 10837, GenBank no. DQ408559), *Periconia igniaria* E.W. Mason and M.B. Ellis (as *Massarina igniaria* (C. Booth) Aptroot, GenBank no. DQ810223), *Byssothecium circinans* Fuckel (CBS 675.92, GenBank no. AY016357) and *Corynespora smithii* (Berk. and Broome) M.B. Ellis (CABI 5649b, GenBank no. GU323201). *Noosia* shared some morphological resemblances to *Conioscypha* Höhn., *Periconiella sensu lato* and *Trichobotrys* Penz. and Sacc., but differed from these genera in many morphological aspects [16]. Consequently, Tanaka et al. [12] resurrected family Periconiaceae in the suborder Massarineae (Pleosporales) and included *Noosia* in this family. However, Tanaka et al. [12] suggested that the taxonomic revision of *Noosia* along with *Periconia* species should be revisited.

In the present study, the LSU nucleotide BLAST result indicated that *Noosia banksiae* is most closely related to *Periconia neobrittanica* Crous strain CPC 37903 (identities = 877/883 bp (99.3%), without gap) and is also similar to *P. homothallica* KT916 (identities = 834/846 bp (98.6%), without gap). Phylogenetic analyses also supported that *Noosia banksiae* has a close relationship with *P. homothallica* KT916 (Figure 1). *Noosia banksiae* can be distinguished from other asexual morphs of *Periconia* species in vivo due to the conidiophores being reduced to conidiogenous cells as well as conidia forming dimorphic characteristics [16]. However, *Noosia banksiae* morphologically resembles the asexual morphs of *Periconia artemisiae*, *P. chimonanthi* and *P. pseudobyssoides* that sporulated on PDA after 3 months in this study (Table 2). Therefore, we synonymize *Noosia banksiae* under *Periconia* as *P. banksiae* based on morphological resemblances and phylogenetic support.

***Periconia byssoides*** Pers., Syn. meth. fung. (Göttingen) 2: 686 (1801) Figure 4.

Index Fungorum number: 144538.

Synonym: *Periconia celtidis* Tennakoon, C.H. Kuo and K.D. Hyde, in Tennakoon, Kuo, Maharachchikumbura, Thambugala, Gentekaki, Phillips, Bhat, Wanasinghe, de Silva, Promputtha and Hyde, Fungal Diversity: 10.1007/s13225-021-00474-w, [35] (2021).

Sexual morph: Undetermined. Asexual morph: Colonies superficial on the host substrate, effuse, hairy, dark brown. Conidiophores 355–635 μm long, 12.5–17 μm wide, macronematous, mononematous, solitary, rarely 1−2 together on stroma, arising from inconspicuous stromatic base, straight or slightly flexuous, 1−3-septate, dark brown to black at the base, paler brown towards the apex, with spherical conidial head, rarely branched at the apex, slightly constricted near conidial head and knot-like near the base, smooth or slightly rough, thick-walled. Conidiogenous cells 6–9 × 5–8 μm ($\bar{x}$ = 7.2 × 6.7 μm, n = 15), polyblastic, discrete, terminal, subglobose to ellipsoidal, light brown, located at nodose apices of conidiophores. Conidia 13–15 × 12.5–14.5 μm ($\bar{x}$ = 14 × 13.6 μm, n = 20), solitary or catenate, globose to subglobose, orangish brown to brown, aseptate, echinulate or verruculose.

Culture characteristics: Conidia germinated on PDA within 18 h. Colonies on PDA fast-growing, reaching 25 diam in one week at room temperature (10–25 °C), medium dense, circular, flattened, dull, surface smooth, with standing white hyphal turfs, edge entire, with well-defined margin, white to yellowish at the middle, white grey at the margin from the above; dark brown to black at the center, light yellowish at the middle, yellowish grey at the margin in reverse. Mycelium immersed in PDA, branched, septate, smooth-walled, brown hyphae.

Material examined: CHINA, Yunnan Province, Kunming City, Kunming Botanical Garden (25°8′32″ N, 102°44′214″ E, altitude 1918 msl), saprobic on dead stems of *Prunus armeniaca* L. (Rosaceae), 7 September 2019, E.F. Yang, DY8 (KUN-HKAS 107383), living culture, KUMCC 20-0264.

Distribution (based on molecular data): China (Taiwan, Yunnan), Japan, Lithuania, Netherlands, Thailand [12,21,24,40,66,67].

Hosts and habitat (based on molecular data): Saprobic on *Ampelopsis brevipedunculata* (Vitaceae) [12], *Angelica sylvestris* (Apiaceae) [24], *Benthamidia japonica* (Cornaceae) [12], *Celtis formosana* (Cannabaceae) [40], *Conium maculatum* (Apiaceae) [24], *Heracleum sosnowskyi* (Apiaceae) [24], *Macaranga tanarius* (Euphorbiaceae) [40], *Magnolia grandiflora* (Magnoliaceae) [21], *Peltophorum* sp. (Fabaceae) [21], *Prunus armeniaca* (this study)*, P. verecunda* (Rosaceae) [12].

Notes: Based on ITS nucleotide BLAST search, the isolate KUMCC 20-0264 is most similar to *Periconia byssoides* H4853 (GenBank no. MW444854) with 99% similarity (535/536 bp). A nucleotide base comparison of the ITS and *tef1-α* sequences between the new isolate KUMCC 20-0264 and other representative strains of *P. byssoides* also revealed nucleotide differences less than 1.5% (Table 3), indicating that the new isolate is conspecific with *P. byssoides* [65]. Phylogenetic analyses demonstrated that the new isolate KUMCC 20-0264 is basal to other *P. byssoides* and *P. celtidis*, with 68% ML and 0.71 PP support (Figure 1) and most closely related with *P. byssoides* strains MFLUCC 18-1548 and MFLUCC 18-1553, which were isolated from decaying pods of *Peltophorum* sp. (Fabaceae) in Thailand [21]. Based on Farr and Rossman [3], *P. byssoides* is reported as a saprobe on *Prunus armeniaca* in Yunnan, China for the first time.

Tennakoon et al. [40] introduced *Periconia celtidis* as a saprobe on dead leaves of *Celtis formosana* and *Macaranga tanarius* from Taiwan. Their phylogenetic analyses showed that *P. celtidis* form a distinct lineage and clustered with *P. byssoides*, *P. pseudobyssoides* and *P. alishanica* with 90% ML and 1.00 PP support (Tennakoon et al. [40] (p. 37)). In the present phylogenetic analyses, *P. celtidis* grouped with other *P. byssoides*, and this result was also supported by PHI results of ITS, LSU, and a combined ITS, LSU, SSU and *tef1-α* sequence dataset (Figure 2). Therefore, we treat *P. celtidis* as a synonym of *P. byssoides* in this study. Morphological characteristics of the new isolate KUMCC 20-0264 are slightly different from other representative isolates of *P. byssoides* and *P. celtidis* in size of conidiophores and conidia which is detailed in Table 4.

***Periconia chimonanthi*** E.F. Yang, H.B. Jiang and Phookamsak, sp. nov. Figure 5.

Index Fungorum number: IF559497.

Etymology: The specific epithet “*chimonanthi*” refers to the host genus, *Chimonanthi*, from which the fungus was collected.

Holotype: KUN-HKAS 107380.

Saprobic on dead stems of *Chimonanthi praecox* (L.) Link. Sexual morph: Undetermined. Asexual morph: Colonies on the substrate superficial, numerous, effuse, dark brown to black, hairy. Conidiophores 410–635 μm long, 8.5–12 μm wide, broader at the base, 14–18 μm wide, macronematous, mononematous, arising usually singly, rarely 2–3 together on stroma, erect or slightly flexuous, swollen at the 3rd cell from the apex, branched at apex, with spherical conidial head, dark brown, septate, smooth to verruculose, thick-walled. Conidiogenous cells 7.5–9.5 × 5–6 μm ($\bar{x}$ = 8.6 × 5.7 μm, n = 20), mono- to polyblastic, brown to dark brown, ovoid to subglobose, terminal, proliferating, with 1–2 conidiogenous loci. Conidia 7–8 × 6–7 μm ($\bar{x}$ = 7.3 × 6.6 μm, n = 20), catenated, globose, initially hyaline greenish-brown, becoming yellowish brown to brown at maturity, aseptate, verruculose. In vitro: sporulated on PDA after 4 months. Mycelium 2–3 μm wide ($\bar{x}$ = 2.5 μm, n = 15), branched, hyaline to brown, with well-developed small, granular, oil droplets, aggregated in hyphal strands. Conidiophores reduced to conidiogenous cells. Conidiogenous cells 7–10 × 4.5–6 μm ($\bar{x}$ = 8.8 × 4.9 μm, n = 20), polyblastic, solitary, erect, lateral and terminal, cylindrical to irregular, luteous to brown, discrete or integrated, determinate, or inconspicuous, percurrent proliferations, with 1–3 conidiogenous loci. Conidia 6–8 × 6–8 μm ($\bar{x}$ = 7.1 × 6.9 μm, n = 20), globose to oblong, or ellipsoidal, subhyaline to brown or dark brown, smooth to verruculose, solitary or in short chains.

Culture characteristics: Conidia germinated within 18–20 h on PDA. Colonies reaching 15 mm diam in one week at 20–25 °C in normal light, dense, circular, flattened to slightly raised, surface slightly rough, with entire edge, floccose to cottony, radially furrowed at the margin, pale greenish grey at the margin, dark greenish towards the center from above and below; not producing pigmentation on medium.

Material examined: CHINA, Yunnan Province, Kunming City, Kunming Institute of Botany (25°8′7″ N, 102°44′17″ E, altitude 1908.6 msl), on dead branches of *Chimonanthi praecox* (Calycanthaceae), 20 July 2019, C.F. Liao, CP05 (KUN-HKAS 107380, holotype), ex-type living culture, KUMCC 20-0266.

Distribution: China (Yunnan).

Host and habitat: Saprobic on dead branches of *Chimonanthi praecox* (Calycanthaceae).

Notes: The nucleotide BLAST result of ITS sequence indicated that *Periconia chimonanthi* sp. nov. (KUMCC 20-0266) is closest to *P. macrospinosa* strain DUCC4111, *Periconia* sp. strain EPU33CB and *Periconia* sp. strain DUCC4151, with 100% similarities and closely related to *Periconia* sp. CY137 (99.82% similarity). Phylogenetic analyses showed that *P. chimonanthi* (KUMCC 20-0266) forms a single branch with *Periconia* sp. CY137 (100% ML, 1.00 PP; Figure 1) and clusters with *P. cortaderiae*, distancing from *P. macrospinosa* (Figure 1). Rodrigues et al. [68] isolated *Periconia* sp. CY137 from the ant’s nest (*Cyphomyrmex wheeleri*) in Central Texas, USA based on culture-dependent methods for studying microfungal diversity in attine gardens. Hence, the proper morphological description and identification of *Periconia* sp. CY137 have not been provided. In our phylogenetic analyses, *Periconia* sp. CY137 was shown to be conspecific with *P. chimonanthi*. Therefore, we identify *Periconia* sp. CY137 as *P. chimonanthi* in this study. *Periconia chimonanthi* can be distinguished from *P. cortaderiae* in having shorter conidiophores (410–635 × 8.5–12 μm vs. 400–800 × 4–9.4 μm), mono- to polyblastic, brown to dark brown, ovoid to subglobose, terminal, proliferating, with 1–2 conidiogenous loci and larger conidia (7–8 × 6–7 μm vs. 4–6.6 × 4.1–7.1 μm). On the other hand, *P. cortaderiae* formed monoblastic conidiogenous cells that were discrete on stipe [19]. A nucleotide base comparison of ITS and *tef1-α* indicated that *P. chimonanthi* differs from *P. cortaderiae* strain MLFUCC 15-0457 (ex-type strain) in 52/526 bp of ITS (9.89%) and 36/830 bp of *tef1-α* (4.34%). Based on Farr and Rossman [3], this is the first report of a *Periconia* species occurring on *Chimonanthi praecox* in Yunnan, China.

***Periconia didymosporum*** (D.Q. Dai and K.D. Hyde) D.Q. Dai and Phookamsak, comb. nov. Figure 6.

Index Fungorum number: IF559498.

Basionym: *Bambusistroma didymosporum* D.Q. Dai and K.D. Hyde, Index Fungorum 225: 1 (2015).

Type information: THAILAND, Chiang Rai Province, Doi Mae Salong, temple side, on decaying culm of bamboo, 15 August 2013, D.-Q. Dai, DDQ00276 (MFLU 15-0057, holotype), ex-type living culture, MFLUCC 13-0862.

Detailed description and illustration: see Adamčík et al. [15].

Notes: *Bambusistroma didymosporum* was introduced by Adamčík et al. [15] as type species of the monotypic genus *Bambusistroma* and treated in the family Massarinaceae. Consequently, Tanaka et al. [12] treated the genus in Periconiaceae. *Bambusistroma didymosporum* resembles the sexual morphs of *Periconia homothallica* and *P. pseudodigitata* in having solitary or grouped ascomata, immersed in a clypeus-like basal stroma, 8-spored, bitunicate, fissitunicate, cylindrical, short pedicellate asci, with well-developed ocular chamber, embedded in anastomosed, cellular pseudoparaphyses, and hyaline, fusiform, 1-septate ascospores, with an entire sheath. *Bambusistroma didymosporum* can be distinguished from *Periconia homothallica* and *P. pseudodigitata* in having larger, subglobose to conical ascomata as well as in having larger asci and ascospores [12,15]. Based on morphological resemblances coupled with phylogenetic analyses, we hence synonymize the genus *Bambusistroma* under *Periconia*, and the new combination *P. didymosporum* is proposed in this study.

The LSU nucleotide BLAST results indicated that *Bambusistroma didymosporum* is highly similar to *Periconia cyperacearum* Crous strain CPC 32138 (identity = 859/875 bp (98%), with two gaps). Phylogenetic analyses demonstrated that *B. didymosporum* clusters with the non-type strain of *Sporidesmium tengii* (HKUCC 10837), distant from *Periconia cyperacearum* strain CPC 32138. *Sporidesmium tengii* was introduced by Wu and Zhuang [68] and is characterized by partly immersed, partly superficial mycelium, composed of branched, septate, brown to dark brown, smooth-walled hyphae, macronematous, mononematous, erect, unbranched, 4–6-septate, brown to dark brown conidiophores, with triangular base, monoblastic, integrated, determinate, terminal, brown, cylindrical, smooth-walled conidiogenous cells without proliferations and holoblastic, acrogenous, solitary, obclavate, 8-septate conidia, tapering towards the apex, brown, paler brown to hyaline at the apex, [68]. The type specimen (WU 1930b) was obtained from unidentified dead leaves in Guangdong Province, China in 1998, and molecular data are unavailable for the type. Shenoy et al. [69] provided molecular data for *Sporidesmium tengii* (HKUCC 10837), which was obtained from the Culture Collection in Novozymes, Beijing, China and indicated that *S. tengii* (HKUCC 10837) is affiliated basal to Melanommataceae, Pleosporales based on the LSU phylogeny. However, morphological characteristics of *S. tengii* (HKUCC 10837) were not examined by Shenoy et al. [69]. Tanaka et al. [12] included *S. tengii* (HKUCC 10837) in Periconiaceae and suggested that the taxonomic status of *S. tengii* and *Noosia* should be re-evaluated in the light of *Periconia* being subdivided into several morphologically similar genera and the polyphyletic status of *Periconia*.

***Periconia pseudobyssoides*** Markovsk. and Kačergius, Mycol. Progr. 13(2): 293 (2013) [2014] Figure 7.

Index Fungorum number: IF804763.

Type information: Lithuania, Vilnius, 54°45′10″ N, 25°16′04″ E, on dead stalks of *Heracleum sosnowskyi*, 28 October 2011, leg. S. Markovskaja, BILAS 50334 (holotype), ex- holotype culture, BILAS 50334 (S1-11P).

Saprobic on dead branch of *Scrophularia ningpoensis* Hemsl. Sexual morph: Undetermined. Asexual morph: Colonies on the substrate superficial, hairy, effuse. Conidiophores 665–755 μm long, 19–21 μm wide, macronematous, mononematous, arising singly on stroma, erect or slightly flexuous, solitary, septate, unbranched, tapering towards the apex, reddish brown to dark brown at the lower part and gradually pale brown toward the apex, with spherical conidial head, smooth to minutely verruculose, thick-walled. Conidiogenous cells 7–9 × 6.5–8 μm ($\bar{x}$ = 8.1 × 7.3 μm, n = 20), mono- to polyblastic, light brown to brown, ovoid to subglobose, terminal, discrete, determinate, proliferating, thick-walled. Conidia 12–14 × 10–13 μm ($\bar{x}$ = 13 × 12 μm, n = 20), catenate in short chains, globose, hyaline to reddish brown at immature stage, becoming brown to dark brown with age, aseptate, verruculose. In vitro: Culture sporulated on PDA after six months and also produced immersed, brown, globose stroma on MEA. Mycelium 3–6 μm wide, septate, pale brown, thick-walled. Conidiophores reduced to conidiogenous cells. Conidiogenous cells 4–6 × 2.5–4 μm ($\bar{x}$ = 5.1 × 3.2 μm, n = 15), monoblastic, solitary, terminal, with small, pimple-like pores, arising from brown and verruculose hyphae. Conidia 13–16 × 12–15 μm ($\bar{x}$ = 15 × 14 μm, n = 20), solitary, globose, reddish-brown, aseptate, verruculose.

Culture characteristics: Conidia germinated within 18–20 h on PDA, colonies on PDA reaching 25 mm diam in one week at 20–25 °C in normal light, medium dense, flattened to raised, circular, surface smooth, with edge entire, cottony to fluffy, colonies from above, cream to pale yellowish at the margin, white in the middle, with granular, packed, pale yellowish at the center, colonies from below, yellowish brown at the margin, brown to dark brown towards the center; not produced pigmentation on PDA.

Material examined: CHINA, Yunnan Province, Chuxiong Yi Autonomous Prefecture, Daguokou Town (24°88′8″ N, 101°16′2″ E, altitude 2016 msl), on dead branch of *Scrophularia ningpoensis* (Scrophulariaceae), 14 August 2019, E.F. Yang, DY6 (KUN-HKAS 107382), living culture, KUMCC 20-0263.

Distribution (based on molecular data): China (Yunnan), Japan, Korea, Lithuania [12,20,24].

Host and habitat (based on molecular data): *Berchemia racemosa* (Rhamnaceae) in terrestrial habitat [12], decaying wood submerged in stream [20], *Heracleum sosnowskyi* (Apiaceae) in terrestrial habitat [24], *Rodgersia podophylla* (Saxifragaceae) in terrestrial habitat [12], *Scrophularia ningpoensis* (Scrophulariaceae) in terrestrial habitat [this study].

Notes: In the ITS and *tef1-a* nucleotide BLAST search, the newly generated strain KUMCC 20-0263 is identical to *Periconia pseudobyssoides* (strain DUCC 0850) with 99.60% and 98.73% similarities and is also closely related to *P. pseudobyssoides* (strains H4151/MAFF 243868, and H 4790/MAFF 243874). Phylogenetic analyses indicated that the generated strain KUMCC 20-0263 clustered with *P. pseudobyssoides* strains DUCC 0850, MAFF 243868 and MAFF 243874 with 58% ML, 0.86 PP support (Figure 1). A nucleotide base comparison of the ITS and *tef1-α* sequences between the new isolate KUMCC 20-0263 and other representative strains of *P. pseudobyssoides* also revealed nucleotide differences of less than 1.5% (Table 5). Moreover, the ITS nucleotide base comparison between KUMCC 20-0263 and the ex-type strain of *P. pseudobyssoides* (BILAS 50334) also revealed the conspecific status of KUMCC 20-0263 with *P. pseudobyssoides* (3/562 bp of ITS (0.53%, <1.5%)). Thus, we identified our newly generated strain KUMCC 20-0263 as *P. pseudobyssoides*, which is reported from *Scrophularia ningpoensis* in Yunnan, China for the first time.

*Periconia pseudobyssoides* was introduced by Markovskaja and Kacergius [24]. The species was found to be a saprobe on dead stalks of *Heracleum sosnowskyi* in Lithuania. *Periconia pseudobyssoides* (KUMCC 20-0263) is morphologically similar to the type of *P. pseudobyssoides* (BILAS 50334) and *P. pseudobyssoides* (DUCC 0850) in producing macronematous, brown, septate, unbranched, verruculose conidiophores, mono- to polyblastic, light brown to brown, ovoid to subglobose, discrete, determinate conidiogenous cells, and globose, brown, aseptate, verruculose conidia, which are borne singly or catenate in short chains. However, these collections vary slightly in size of conidiophores and conidia, likely depending on environmental factors and host associations [20,24].

***Periconia thysanolaenae*** E.F. Yang, H.B. Jiang and Phookamsak, sp. nov. Figure 8.

Index Fungorum number: IF559499.

Etymology: The specific epithet “*thysanolaenae*” refers to the host genus, *Thysanolaena*, from which the species was collected.

Holotype: KUN-HKAS 107381.

Saprobic on dead culms of *Thysanolaena latifolia* (Roxb. ex Hornem.) Honda. Sexual morph: Undetermined. Asexual morph: Colonies on the substrate, superficial, numerous, hairy, effuse. Conidiophores 425–550 μm long, 8–13.5 μm wide, 16.5–22.5 μm wide at the base, macronematous, mononematous, erect or slightly flexuous, solitary, septate, reddish brown to dark brown, and gradually pale brown toward the apex, smooth to minutely verruculose, thick-walled, branched at the apex (the apex of ultimate cells of the primary conidiophores proliferate, producing 2–3 secondary conidiophores). Conidiogenous cells 5.5–7 × 4–5 μm ($\bar{x}$ = 6 × 4.4 μm, n = 20), polyblastic, ovoid to subglobose, light brown to brown, terminal, proliferating. Conidia 4.5–6 × 4–6 μm ($\bar{x}$ = 5.3 × 5 μm, n = 20), catenate, globose, hyaline to reddish brown at immature stage, becoming brown to dark brown at maturity, aseptate, rough-walled, verruculose.

Culture characteristics: Conidia germinated within 18–20 h on PDA. Colonies growing on PDA reaching 25 mm diam in one week at 20–25 °C, colonies moderately dense, circular, flattened, surface smooth, with edge entire, forming white to grey mycelial turfs on surface with age, velvety to fluffy, colonies from above white greyish to pale brown, yellowish brown, with small black dots in reverse; producing yellowish to greenish pigmentation in PDA.

Material examined: CHINA, Yunnan Province, Xishuangbanna Dai Autonomous Prefecture, Mengla County, Bubeng Village, on dead culms of *Thysanolaena latifolia* (Poaceae), 23 January 2019, E.F. Yang, DY5 (KUN-HKAS 107381, holotype), ex-type living culture, KUMCC 20-0262.

Distribution: China (Yunnan).

Host and habitat: Saprobic on *Thysanolaena latifolia* in terrestrial habitat.

Notes: Results from the ITS nucleotide BLAST search indicated that the newly generated strain KUMCC 20-0262 is most similar to *Periconia minutissima* strain MUT 2887 (GenBank. no. MG813227) with 98.73% similarity and is similar to *P. macrospinosa* isolate UFMG PEZ8 (GenBank no. KY364630) with 98.41% similarity. Phylogenetic analyses demonstrated that KUMCC 20-0262 is sister to *P. minutissima* with 67% ML, 0.80 PP support and clustered with *P. homothallica* (KT 916) and *P. banksiae* (CBS 129526) with low support, distancing from *P. macrospinosa* (Figure 1). Based on a nucleotide pairwise comparison of ITS and *tef1-α* sequences, *P. thysanolaenae* sp. nov. (KUMCC 20-0262) differs from *P. minutissima* strain MUT 2887 in 5/458 bp of ITS (1.09%), differs from *P. homothallica* strain KT 916 (type strain) in 25/530 bp of ITS (4.72%), and differs from *P. banksiae* strain CBS 129526 (type strain) in 48/602 bp of ITS (7.97%). *Periconia minutissima* strain MUT 2887 formed a distinct clade with *P. minutissima* strain MFLUCC 15-0245 in this study. Bovio et al. [70] isolated *P. minutissima* strain MUT 2887 from *Sycon ciliatum* in the Atlantic Ocean using a culture method and identified the strain as *P. minutissima* based on a high percentage of homologies, with sequences available in GenBank. However, *P. minutissima* strain MFLUCC 15-0245 remains unpublished. Our new isolate KUMCC 20-0262 is not significantly different from *P. minutissima* strain MUT 2887 based on the ITS nucleotide base comparison (1.09%); however, *P. minutissima* strain MUT 2887 has no morphological support for species identification. Unfortunately, the ex-type strain of *P. minutissima* is unavailable. We, therefore, introduced our new collection as *P. thysanolaenae* sp. nov.

Morphological characteristics of *Periconia thysanolaenae* align with described *Periconia* species in having erect or flexuous, faint to reddish brown, or dark brown, conidiophores, with spherical conidial head at the apex, polyblastic, ovoid to subglobose, light brown to brown, terminal conidiogenous cells, and catenate, globose, and brown to dark brown aseptate conidia. However, *P. thysanolaenae* can be distinguished from other phylogenetically related species in having proliferating, 2–3 secondary conidiophores. *Periconia minutissima* is characterized by effuse, brown, colonies, with dark brown, loose, very slender, few septate, translucent conidiophores, divided above into very short branches, with botrytis-like apex and globose, colorless conidia [71], while *P. homothallica* is represented by its sexual morph, and *P. banksiae* formed an asexual morph in vitro that lack conspicuous conidiophores [12,16].

## 4. Discussion

Tanaka et al. [12] resurrected the family Periconiaceae in the suborder Massarineae (Pleosporales, Dothideomycetes) based on morphological and phylogenetic approaches and updated the taxonomic treatment of Periconiaceae in modern fungal systematics. They accepted *Bambusistroma*, *Flavomyces*, *Noosia* and *Periconia* in the family, while a single lineage of “*Sporidesmium tengii*” was also included in Periconiaceae [12]. However, *Bambusistroma*, *Flavomyces* and *Noosia* represented as monotypic genera always grouped with *Periconia* species. Moreover, *Periconia* species did not form a single clade within Periconiaceae, indicating that *Periconia* may be polyphyletic [2,12,18,19,20,21,22,40]. Thus, Tanaka et al. [12] suggested that *Periconia* probably can be subdivided into several morphologically similar genera or that the taxonomic status of the monotypic genera and “*Sporidesmium tengii*” should be re-evaluated.

In this study, we re-evaluate the taxonomic status of *Bambusistroma* and *Noosia* based on morphological and multigene phylogenetic approaches. Morphological comparison of *Bambusistroma* and the sexual morphs of *Periconia* species (viz. *P. homothallica* and *P. pseudodigitata*) showed that *Bambusistroma* has similar morphology with the sexual morph of *Periconia* species (see notes of *P. didymosporum*), while *P. artemisiae*, *P. chimonanthi* and *P. pseudobyssoides* sporulated in vitro are not significantly different from *Noosia* (see notes of *P. banksiae*). Therefore, these two monotypic genera are treated as synonyms of *Periconia* in this study. Knapp et al. [4] introduced a monotypic genus *Flavomyces* to accommodate the root endophytic species, *F. fueloephazae* (as *F. fulophazii*), isolated from the root of *Festuca vaginata* in Hungary based on culturable and phylogenetic approaches. Morphological characteristics of *Flavomyces* are undescribed. In this study, *F. fueloephazae* formed an independent clade basal to *Periconia prolifica* Anastasiou, *P. citlaltepetlensis* Calvillo, Cobos-Villagrán and Raymundo, *P. igniaria*, *P. epilithographicola* Coronado-Ruiz et al., *P. caespitosa* Cantillo, Gusmão and Madrid, *P. variicolor* S.A. Cantrell, Hanlin and E. Silva, *P. minutissima* and *P. macrospinosa* with significant support in ML analyses (73% ML; Figure 1). Since morphology is poorly known, the generic status of *Flavomyces* may need to be clarified.

Phylogenetic relationships of many *Periconia* species are not well-resolved, such as the relationships of *P. cookei* E.W. Mason and M.B. Ellis, *P. delonicis* Jayasiri, E.B.G. Jones and K.D. Hyde, *P. elaeidis* T. Sunpapao and K.D. Hyde, *P. palmicola* J.F. Li and Phookamsak and *P. verrucosa* Phukhams. et al. These species are grouped together with significant support in ML analyses (98%; Figure 1); however, the interspecific relationships of these species could not be resolved. These species may be conspecific and contain a high degree of genetic variation. The conspecific status of these ambiguous species should be re-evaluated. Moreover, phylogenetic relationships of *P. byssoides*, *P. celtidis*, *P. pseudobyssoides* and *P. salina* are not well-resolved in this study and it is in agreement with previous studies [18,51,72]. We re-evaluated the conspecific status of *P. byssoides*, *P. celtidis*, *P. pseudobyssoides* and *P. salina* based on the pairwise homoplasy index (PHI) test. The PHI test of the individual gene (ITS, LSU, and *tef1-α*) and the combined ITS, LSU, SSU and *tef1-α* sequence dataset resulted in no significant recombination among *P. byssoides*, *P. pseudobyssoides* and *P. salina* (Figure 2), indicating that *P. byssoides*, *P. pseudobyssoides* and *P. salina* are not conspecific. In contrast, the PHI test resulted in significant recombination between *P. byssoides* and *P. celtidis*, and polymorphic nucleotide comparison between *P. byssoides* and *P. celtidis* (Table 2) also supported that *P. celtidis* is conspecific with *P. byssoides*. Hence, *P. celtidis* is synonymized under *P. byssoides* in this study.

Out of 121 morphologically accepted species, only 28 species (including three new species and two novel combinations provided in this study) have molecular data to clarify their phylogenetic placements in Periconiaceae [2,40]. Most available sequences of *Periconia* species are limited to ITS and LSU gene regions. Sixteen species have sequence data of a protein-coding gene (*tef1-α*), but most species do not have useful phylogenetic markers (i.e., mtSSU, *rpb1*, *rpb2* and *tub2*) to identify them to the species level in Periconiaceae. Moreover, some *Periconia* species show an intraspecific variation, such as *P. byssoides*, *P. cortaderiae*, *P. prolifica* and *P. pseudobyssoides*. More reliable phylogenetic markers may be required to support their conspecific status. In addition, two strains of *P. minutissima* (MUT 2887 and MFLUCC 15-0245) and two strains of *P. igniaria* (CBS 379.86 and CBS 485.96) form a distinct clade separated from each other within Periconiaceae. These strains are not represented by the ex-type strains. Therefore, the phylogenetic status of *P. minutissima* and *P. igniaria* remain in doubt. The epitypification of *P. minutissima* and *P. igniaria* are in urgent need of clarification regarding their phylogenetic status within Periconiaceae.

Nevertheless, the type specimens of *Periconia lichenoides* have been lost while, the DNA sequence data of this type species are also unavailable in GenBank. Tanaka et al. [12] judged *Periconia* to be a member of Dothideomycetes due to the morphological resemblance of *Periconia sensu stricto* (e.g., *P. byssoides*, *P. cookei*, *P. igniaria*, and *P. digitata*) with *P. lichenoides*. The taxonomic treatment proposed by Tanaka et al. [12] was followed by various subsequent authors [2,6,17,18,19,20,21,22,43,51,52,72]. However, this taxonomic status is somewhat questionable due to the lack of information from the type species of *Periconia*. Neotypification or designation of reference specimens of *P. lichenoides* is urgently needed to clarify the taxonomic status of *Periconia* in Dothideomycetes.

## Figures and Tables

**Figure 1 jof-08-00243-f001:**
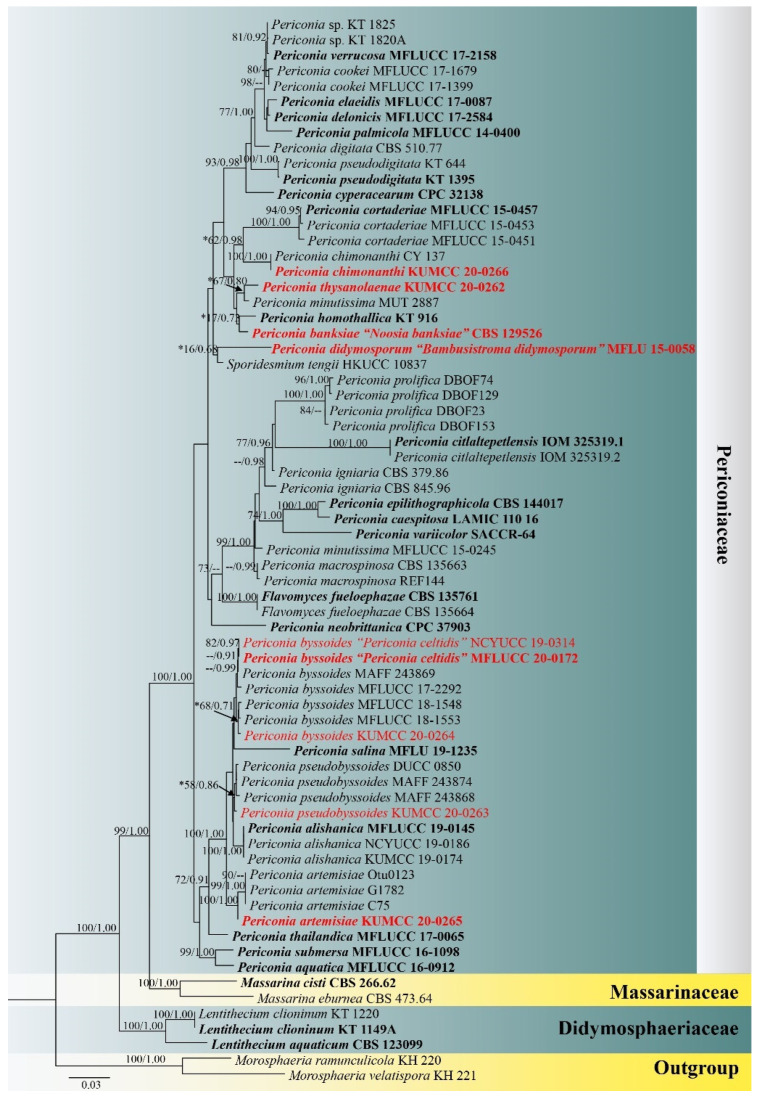
Phylogram generated from maximum likelihood analysis based on combined ITS, LSU, SSU and *tef1-α* sequence data. Bootstrap values for maximum likelihood equal to or greater than 70% and Bayesian posterior probabilities equal to or greater than 0.95 are placed above the nodes as ML/PP. An asterisk (*) indicates insignificant support values corresponding with the newly generated strains and discussed species in this study. Ex-type strains are in bold, and the newly generated species and combination species are indicated in red.

**Figure 2 jof-08-00243-f002:**
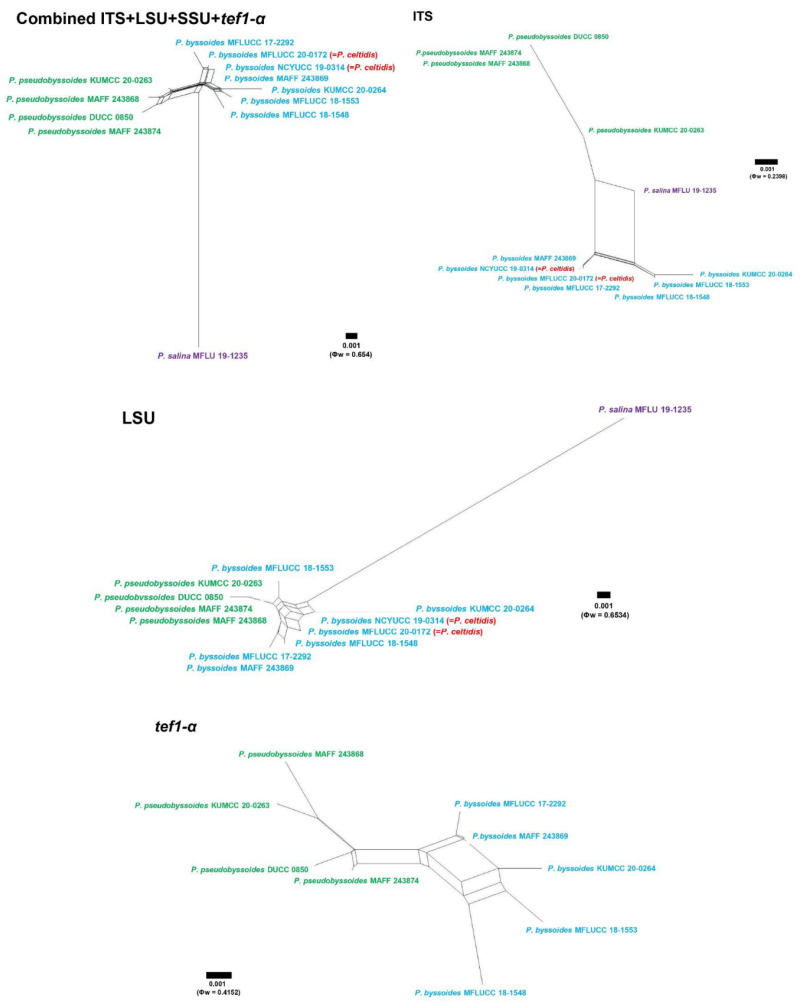
Split graphs showing the results of the pairwise homoplasy index (PHI) tests of closely related taxa using LogDet transformation and split decomposition. PHI test results (Φw) ≤ 0.05 indicate significant recombination within the dataset.

**Figure 3 jof-08-00243-f003:**
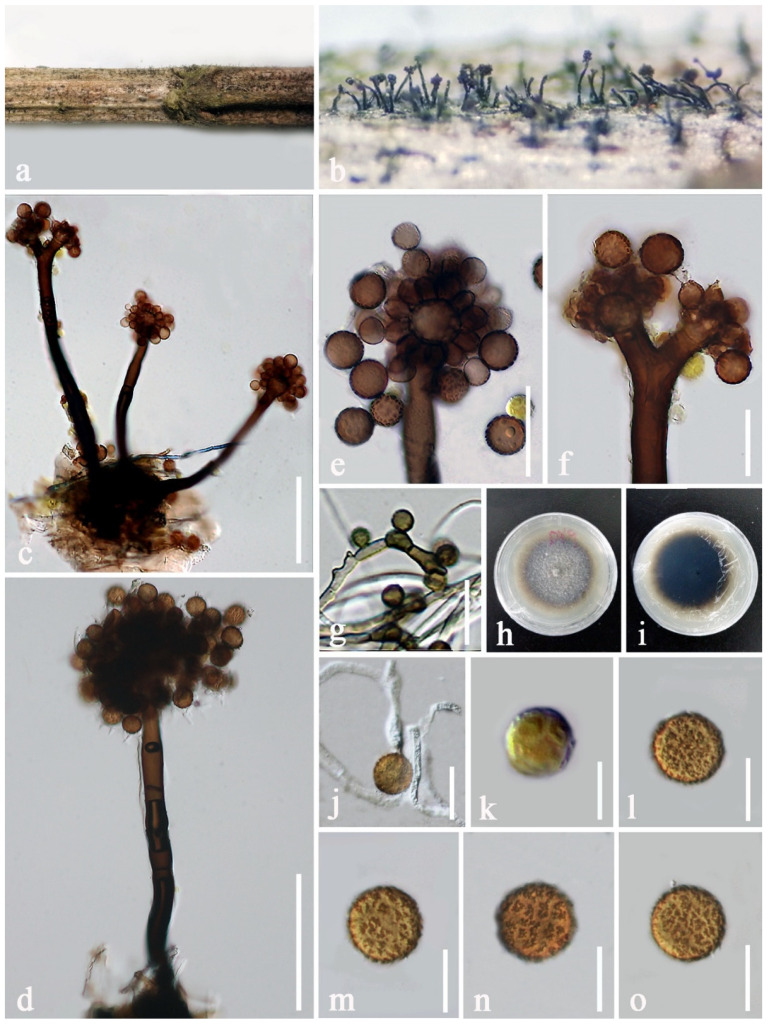
*Periconia artemisiae* (KUN-HKAS 107384, holotype). (**a**) Colonies on the host substrate; (**b**) close-up of colonies on substrate; (**c**,**d**) conidiophores; (**e**,**f**) conidial heads bearing conidiogenous cells and conidia. (**g**) conidia sporulated on PDA after three months. (**h**,**i**) colony on PDA ((**h**) = from above, (**i**) = from below)); (**j**) germinating conidium; (**k**–**o**) conidia. Scale bars: (**c**,**d**) = 100 μm, (**e**,**f**) = 30 μm, (**j**) = 20 μm, g, (**l**–**o**) = 15 μm, (**k**) = 10 μm.

**Figure 4 jof-08-00243-f004:**
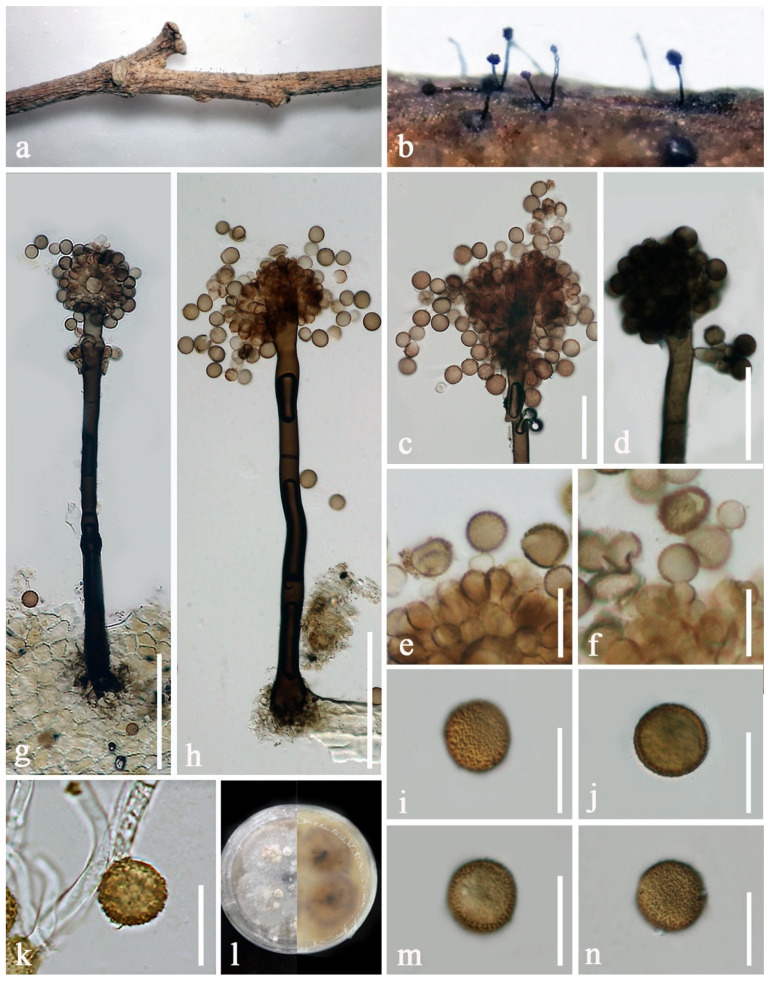
*Periconia byssoides* (KUN-HKAS 107383). (**a**) Colonies on the host substrate; (**b**) closed up conidiophores on host; (**c**,**d**) conidial heads bearing conidiogenous cells and conidia; (**e**,**f**) conidiogenous cells with attached conidia; (**g**,**h**) conidiophores with spherical conidial heads; (**i**,**j**,**m**,**n**) conidia; (**k**) germinated conidia; (**l**) forward and reverse colonies on PDA. Scale bars: (**g**,**h**) = 100 μm, (**c**,**d**) = 50 μm, (**e**,**f**) = 20 μm, (**i**–**k**,**m**,**n**) = 15 μm.

**Figure 5 jof-08-00243-f005:**
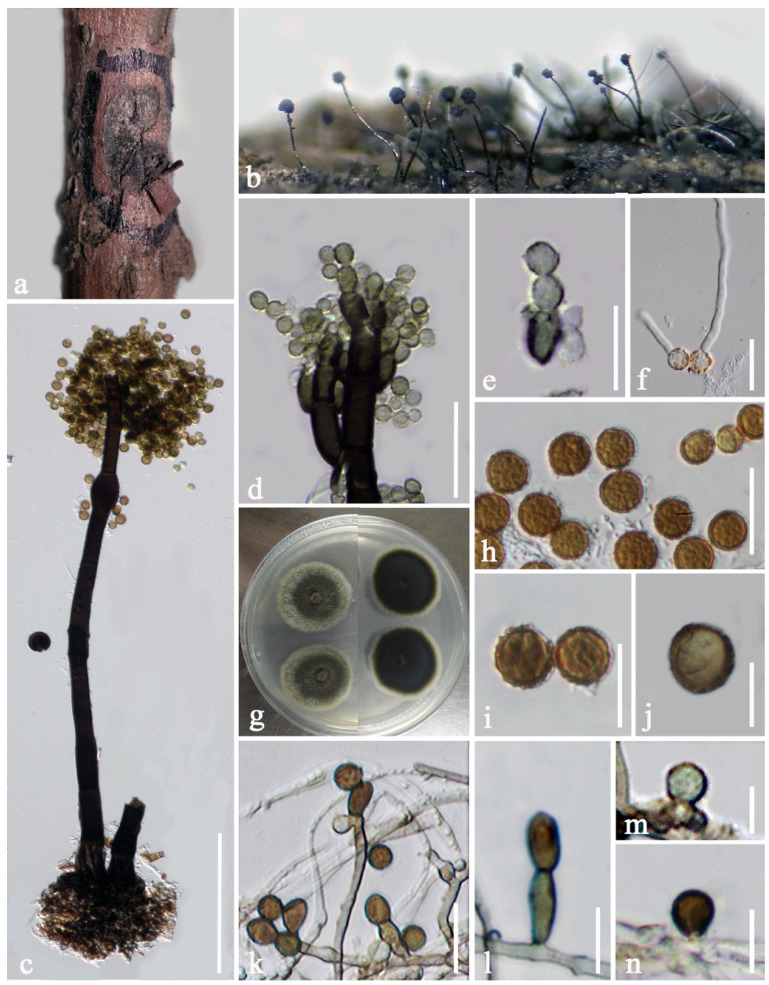
*Periconia chimonanthi* (KUN-HKAS 107380, holotype). (**a**) Located fungal colonies on host substrate; (**b**) closed up of conidiophores on host surface; (**c**) conidiophore with spherical conidial head; (**d**,**e**) conidiogenous cells bearing conidia in short chains; (**f**) germinated conidium; (**g**) colony from above and below; (**h**–**j**) conidia; (**k**–**n**) conidiogenous cells bearing conidia sporulated in vitro. Scale bars: (**c**) = 100 μm, (**d**) = 30 μm, (**f**) = 20 μm, (**e**,**h**,**k**) =15 μm, (**i**,**j**,**l**,**n**) = 10 μm, (**m**) = 5 μm.

**Figure 6 jof-08-00243-f006:**
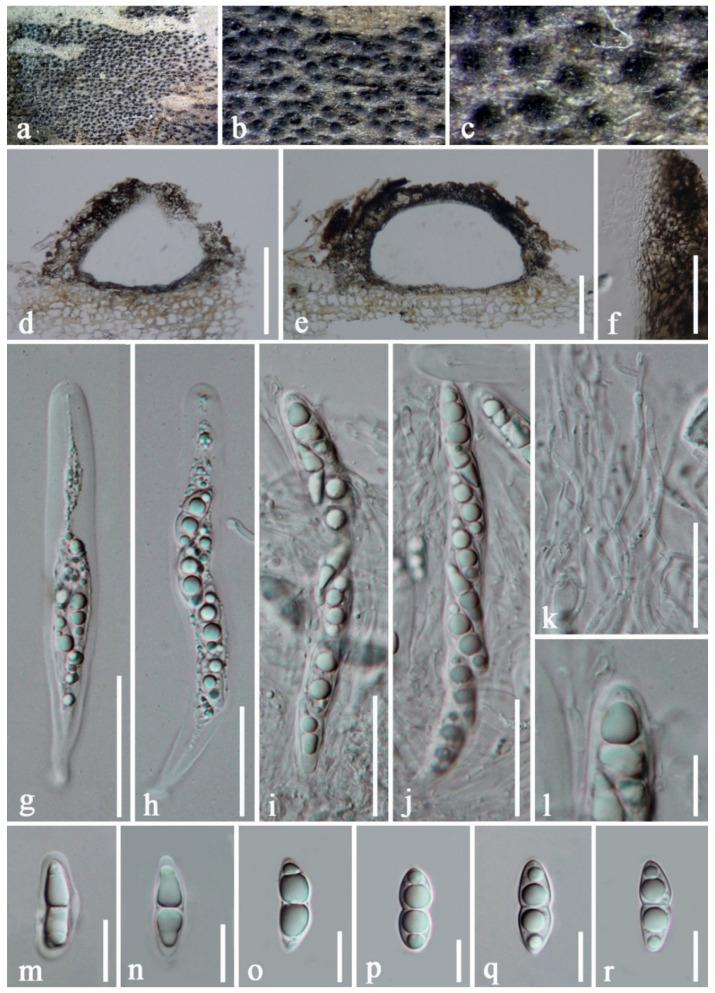
*Periconia didymosporum* (MFLU 15-0057, holotype of *Bambusistroma didymosporum*). (**a**–**c**) Ascomata developing on bamboo culm; (**d**,**e**) vertically section of ascomata; (**f**) section through peridium; (**g**–**j**) immature and mature asci; (**k**) pseudoparaphyses; (**l**) an apical ascus; (**m**–**r**) ascospores. Scale bars: (**d**,**e**) = 100 μm, (**f**–**k**) = 30 μm, (**l**–**n**) = 10 μm. Photo credit by D.Q. Dai.

**Figure 7 jof-08-00243-f007:**
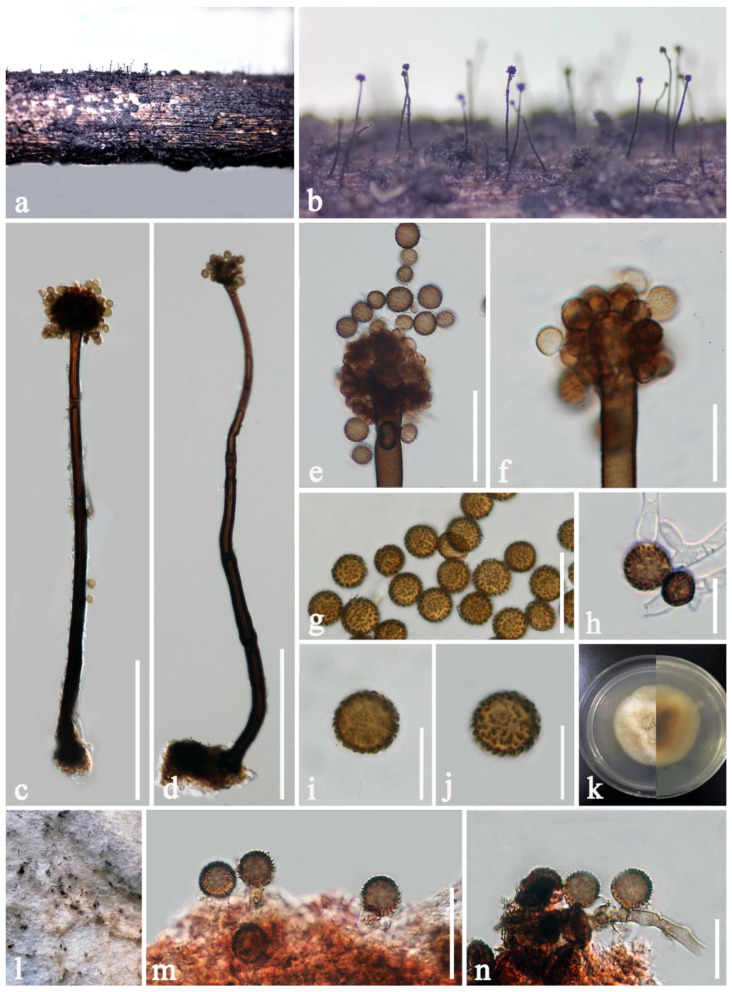
*Periconia pseudobyssoides* (KUN-HKAS 107382). (**a**) Standing hairy-like conidiophores of *P. pseudobyssoides* on host substrate; (**b**) close-up of conidiophores on substrate (**c**,**d**) conidiophores bearing spherical, conidial heads; (**e**,**f**) spherical conidial heads bearing conidiogenous cells and conidia; (**g**,**i**,**j**) conidia; (**h**) germinated conidia; (**k**) colony on PDA from above and below; (**l**) close-up conidial masses on PDA; (**m**,**n**) conidia sporulated in vitro. Scale bars: (**c**,**d**) = 200 μm, (**e**) = 50 μm, (**g**,**m**) = 30 μm, (**f**,**n**) = 20 μm, (**h**–**j**) = 15 μm.

**Figure 8 jof-08-00243-f008:**
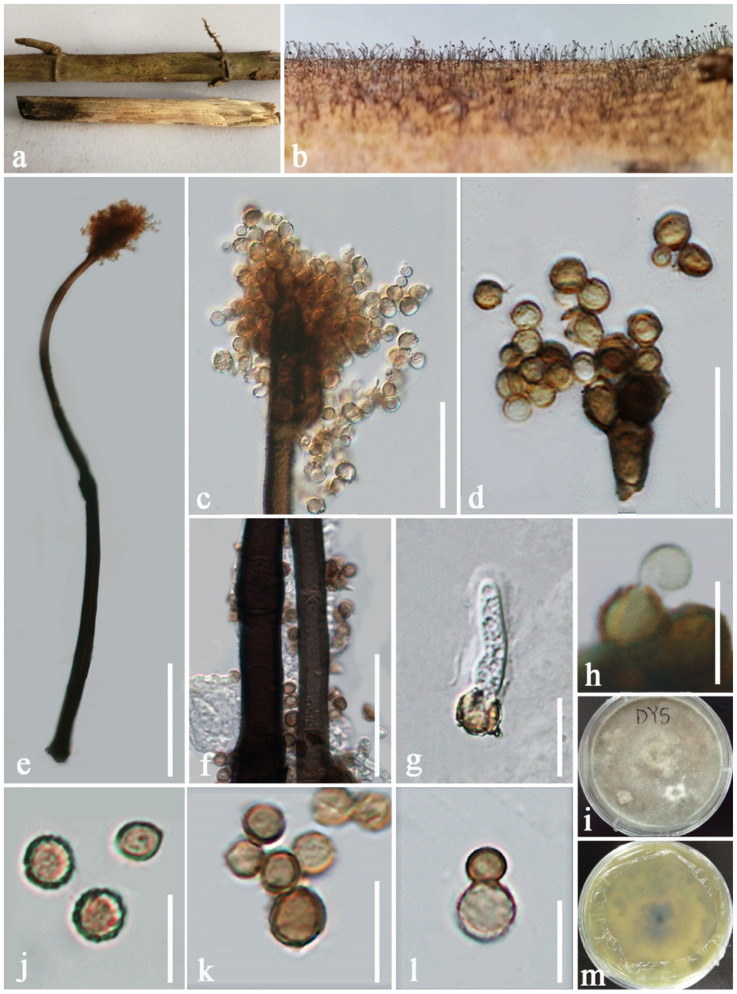
*Periconia thysanolaenae* (KUN-HKAS 107381, holotype) (**a**) Dead culms of *Thysanolaena latifolia* (Poaceae); (**b**) close-up of conidiophores on host substrate; (**c**) apical branch conidiophores with conidial head; (d) polyblastic conidiogenous cells bearing catenate conidia; (**e**) a conidiophore; (**f**) close-up base of the conidiophores; (**g**) germinated conidium; (**h**) conidiogenous cell with developing conidium on upper fertile part; (**j**) single conidia; (**k**,**l**) catenate conidia; **(i**,**m**) colony from above and below. Scale bars: (**e**) = 100 μm, (**c**) = 30 μm, (**f**,**d**) = 20 μm, (**j**) = 10 μm, (**g**,**h**,**k**,**l**) = 5 μm.

**Table 1 jof-08-00243-t001:** Species names, strain numbers and corresponding GenBank accession numbers of taxa used in the present phylogenetic analyses.

Species Name	Strain/Voucher No.	GenBank Accession Numbers			
		ITS	LSU	SSU	*tef1*-α
*Flavomyces fulophazae*	CBS 135664	KP184000	KP184039	KP184081	/
** *Flavomyces fulophazae* **	**CBS 135761**	**NR_137960**	**NG_058131**	**NG_061191**	**/**
** *Lentithecium aquaticum* **	**CBS 123099**	**NR_160229**	**NG_064211**	**NG_016507**	**GU349068**
** *Lentithecium clioninum* **	**KT 1149A**	**LC014566**	**AB807540**	**AB797250**	**AB808515**
*Lentithecium clioninum*	KT 1220	LC014567	AB807541	AB797251	AB808516
** *Massarina cisti* **	**CBS 266.62**	**/**	**AB807539**	**AB797249**	**AB808514**
*Massarina eburnea*	CBS 473.64	/	GU301840	GU296170	GU349040
*Morosphaeria ramunculicola*	KH 220	/	AB807554	AB797264	AB808530
*Morosphaeria velatispora*	KH 221	LC014572	AB807556	AB797266	AB808532
*Periconia alishanica*	KUMCC 19-0174	MW063167	MW063231	/	MW183792
** *Periconia alishanica* **	**MFLUCC 19-0145**	**MW063165**	**MW063229**	**/**	**MW183790**
*Periconia alishanica*	NCYUCC 19-0186	MW063166	MW063230	/	MW183791
** *Periconia aquatica* **	**MFLUCC 16-0912**	**KY794701**	**KY794705**	**/**	**KY814760**
** * Periconia artemisiae * **	** KUMCC 20-0265 **	** MW448657 **	** MW448571 **	** MW448658 **	** MW460898 **
***Periconia banksiae*** (as *Noosia banksiae*)	**CBS 129526**	**/**	**NG_064279**	**/**	**/**
* Periconia byssoides *	KUMCC 20-0264	MW444854	MW444855	MW444856	MW460895
*Periconia byssoides*	MAFF 243869	LC014582	AB807569	AB797279	AB808545
*Periconia byssoides*	MFLUCC 17-2292	MK347751	MK347968	MK347858	MK360069
*Periconia byssoides*	MFLUCC 18-1548	MK347794	MK348013	MK347902	MK360070
*Periconia byssoides*	MFLUCC 18-1553	MK347806	MK348025	MK347914	MK360068
***Periconia byssoides*** (as *Periconia celtidis*)	**MFLUCC 20-0172**	**MW063162**	**MW063226**	**/**	**/**
*Periconia byssoides* (as *Periconia celtidis*)	NCYUCC 19-0314	MW063163	MW063227	/	/
** *Periconia caespitosa* **	**LAMIC 110 16**	**MH051906**	**MH051907**	**/**	**/**
** * Periconia chimonanthi * **	** KUMCC 20-0266 **	** MW448660 **	** MW448572 **	** MW448656 **	** MW460897 **
** *Periconia citlaltepetlensis* **	**IOM 325319.1**	**MH890645**	**MT625978**	**/**	**/**
*Periconia citlaltepetlensis*	IOM 325319.2	MT649221	MT649216	/	/
*Periconia cookei*	MFLUCC 17-1399	MG333490	MG333493	/	MG438279
*Periconia cookei*	MFLUCC 17-1679	/	MG333492	/	MG438278
*Periconia cortaderiae*	MFLUCC 15-0451	KX965734	KX954403	KX986346	KY429208
*Periconia cortaderiae*	MFLUCC 15-0453	KX965733	KX954402	/	KY320574
** *Periconia cortaderiae* **	**MFLUCC 15-0457**	**KX965732**	**KX954401**	**KX986345**	**KY310703**
** *Periconia cyperacearum* **	**CPC 32138**	**NR_160357**	**NG_064549**	**/**	**/**
** *Periconia delonicis* **	**MFLUCC 17-2584**	**/**	**NG_068611**	**NG_065770**	**MK360071**
***Periconia didymosporum*** (as *Bambusistroma didymosporum*)	**MFLU 15-0058**	**KP761734**	**KP761731**	**KP761738**	**KP761728**
*Periconia digitata*	CBS 510.77	LC014584	AB807561	AB797271	AB808537
** *Periconia elaeidis* **	**MFLUCC 17-0087**	**MG742713**	**MH108552**	**MH108551**	**/**
** *Periconia epilithographicola* **	**CBS 144017**	**NR_157477**	**/**	**/**	**/**
** *Periconia homothallica* **	**KT 916**	**AB809645**	**AB807565**	**AB797275**	**/**
*Periconia igniaria*	CBS 379.86	LC014585	AB807566	AB797276	AB808542
*Periconia igniaria*	CBS 845.96	LC014586	AB807567	AB797277	AB808543
*Periconia macrospinosa*	CBS 135663	KP183999	KP184038	KP184080	/
*Periconia macrospinosa*	REF144	JN859364	JN859484	/	/
*Periconia minutissima*	MFLUCC 15-0245	KY794703	KY794707	/	/
*Periconia minutissima*	MUT 2887	MG813227	/	/	/
** *Periconia neobrittanica* **	**CPC 37903**	**NR_166344**	**NG_068342**	**/**	**/**
** *Periconia palmicola* **	**MFLUCC 14-0400**	**/**	**NG_068917**	**MN648319**	**MN821070**
*Periconia prolifica*	DBOF23	JQ724384	/	/	/
*Periconia prolifica*	DBOF74	JQ724435	/	/	/
*Periconia prolifica*	DBOF129	JQ724490	/	/	/
*Periconia prolifica*	DBOF153	JQ724513	/	/	/
*Periconia pseudobyssoides*	DUCC 0850	MG333491	MG333494	/	MG438280
* Periconia pseudobyssoides *	KUMCC 20-0263	MW444851	MW444852	MW444853	MW460894
*Periconia pseudobyssoides*	MAFF 243868	LC014587	AB807568	AB797278	AB808544
*Periconia pseudobyssoides*	MAFF 243874	LC014588	AB807560	AB797270	AB808536
*Periconia pseudodigitata*	KT 644	LC014589	AB807562	AB797272	AB808538
** *Periconia pseudodigitata* **	**KT 1395**	**NR_153490**	**NG_059396**	**NG_064850**	**AB808540**
** *Periconia salina* **	**MFLU 19-1235**	**MN047086**	**MN017846**	**MN017912**	**/**
*Periconia* sp.	C75	MK304380	/	/	/
*Periconia* sp.	CY 137	HQ607981	/	/	/
*Periconia* sp.	G1782	MK247789	/	/	/
*Pericona* sp.	KT 1825	/	AB807573	AB797283	AB808549
*Pericona* sp.	KT 1820A	/	AB807572	AB797282	AB808548
*Pericona* sp.	Out0123	MT908499	/	/	/
** *Periconia submersa* **	**MFLUCC 16-1098**	**KY794702**	**KY794706**	**/**	**KY814761**
** *Periconia thailandica* **	**MFLUCC 17-0065**	**KY753887**	**KY753888**	**KY753889**	**/**
** * Periconia thysanolaenae * **	** KUMCC 20-0262 **	** MW442967 **	** MW444850 **	** MW448659 **	** MW460896 **
** *Periconia variicolor* **	**SACCR-64**	**DQ336713**	**/**	**/**	**/**
** *Periconia verrucosa* **	**MFLUCC 17-2158**	**MT310617**	**MT214572**	**MT226686**	**MT394631**
** *Sporidesmium tengii* **	**HKUCC 10837**	**/**	**DQ408559**	**/**	**/**

* The newly generated sequences are indicated in red, and the ex-type strains are in bold. **Abbreviations: CBS**: Culture Collection of the Westerdijk Fungal Biodiversity Institute, Utrecht, Netherlands; **CPC**: Culture Collection of Pedro Crous, Netherlands; **DBOF**: DNA Barcoding Ocean Fungi; **DUCC**: Dali University Culture Collection, Yunnan, China; **HKUCC**: The University of Hong Kong Culture Collection, Hong Kong, China; **IOM**: Instituto de Oftalmología “Fundación Conde de Valenciana” IAP Mexico Culture Collection; **KH**: K. Hirayama; **KT**: Kaz. Tanaka; **KUMCC**: Kunming Institute of Botany Culture Collection, Kunming, China; **LAMIC**: Laboratorio de Micologia, Universidade Estadual de Feira de Santana, Brazil; **MAFF**: Ministry of Agriculture, Forestry and Fisheries, Japan; **MFLU**: Herbarium of Mae Fah Luang University, Chiang Rai, Thailand; **MFLUCC**: Mae Fah Luang University Culture Collection, Chiang Rai, Thailand; **MUT**: Mycotheca Universitatis Taurinensis, Department of Life Sciences and Systems Biology, University of Turin, Turin, Italy; **NCYUCC**: National Chiayi University Culture Collection, Taiwan.

**Table 2 jof-08-00243-t002:** Synopsis of *Periconia artemisiae*, *P.*
*banksiae* (≡ *Noosia banksiae*), *P. chimonanthi* and *P. pseudobyssoides*.

Species Name	Morphological Characteristics In Vitro	
	Conidiogenous Cells	Conidia
*Periconia artemisiae* (KUN-HKAS 107384)	2.5–4 × 1–2 μm, mono- to polyblastic, with 1–2 conidiogenous loci, lateral, or integrated, lateral and terminal, brown to dark brown, inconspicuous, giving rise to solitary conidia, or in short chains.	5–8 × 4–5 μm globose, brown to dark brown, aseptate, smooth to slightly verruculose.
*Periconia banksiae* (≡ *Noosia banksiae*, CBS H-20587)	Solitary, lateral, or integrated, inconspicuous, lateral and terminal, with small, pimple-like pores of up to 0.5 µm diam.	Dimorphic: primary conidia (4–)7–10(–13) × (3.5–)4(–5) µm, aseptate, globose to fusoid-ellipsoidal, subhyaline to brown, smooth to verruculose with age, solitary or in short, branched chains. Secondary conidia 5–15 × 4–5 µm, phragmosporous, brown, verruculose, arising from disarticulating hyphal cells, initially in short chains, forming directly on conidiogenous cells when mature.
*P. chimonanthi* (KUN-HKAS 107380)	7–10 × 4.5–6 μm, polyblastic, solitary, erect, lateral and terminal, cylindrical to irregular, luteous to brown, discrete or integrated, determinate, or inconspicuous, percurrent proliferations, with 1–3 conidiogenous loci.	6–8 × 6–8 μm, globose to oblong, or ellipsoidal, subhyaline to brown or dark brown, smooth to verruculose, solitary or in short chains.
*P. pseudobyssoides* (KUN-HKAS 107382)	4–6 × 2.5–4 μm, monoblastic, solitary, terminal, inconspicuous, with small, pimple-like pores, arising from brown and verruculose hyphae.	13–16 × 12–15 μm, solitary, globose, reddish-brown, aseptate, verruculose.

Notes: Conidiophores reduced to conidiogenous cells for all of these species. The morphological characteristics of *Periconia banksiae* (≡ *Noosia banksiae*) are adopted from Crous et al. [16].

**Table 3 jof-08-00243-t003:** Polymorphic nucleotides from the LSU, ITS and *tef1-α* sequence data of *Periconia byssoides* and *P. celtidis*.

Species Name	Strain Number	Nucleotide Base Differences																					
		LSU (Base Position)						ITS (Base Position)				*tef1-α* (Base Position)											
		361	431	432	468	485	633	42	48	459	495	69	105	237	309	328	336	426	435	567	660	723	915
*Periconia byssoides*	KUMCC 20-0264	T	G	A	A	T	C	-	T	T	T	T	T	T	T	G	T	G	C	T	T	G	T
*Periconia byssoides*	MAFF 243869	T	T	T	G	T	C	-	C	C	C	T	C	T	T	G	C	G	T	T	T	G	T
*Periconia byssoides*	MFLUCC 17-2292	T	T	T	A	T	T	-	C	C	C	T	C	T	T	G	C	G	T	T	G	G	T
*Periconia byssoides*	MFLUCC 18-1548	T	G	A	A	T	C	-	T	T	C	T	T	C	C	A	T	G	T	C	T	A	C
*Periconia byssoides*	MFLUCC 18-1553	C	G	A	A	T	C	A	T	T	C	C	T	T	T	G	T	A	T	C	T	G	T
*Periconia celtidis*	MFLUCC 20-0172	T	T	T	G	C	C	-	C	C	C	Sequence unavailable											
*Periconia celtidis*	NCYUCC 19-0314	T	T	T	G	C	C	-	C	C	C	Sequence unavailable											

**Table 4 jof-08-00243-t004:** Morphological comparison of *Periconia byssoides* strains isolated from different hosts based on available molecular data.

Taxa	Morphological Characteristics			References
*Periconia byssoides* Strain No.	Conidiophores	Conidiogenous Cells	Conidia	
KUMCC 20-0264	355–635 × 12.5–17 μm, macronematous, mononematous, solitary, rarely 1−2 together on stroma, 1−3-septate, dark brown to black at the base, paler brown towards the apex, unbranched, rarely branched at the apex, knot-like near the base, smooth or slightly rough, thick-walled.	6–9 × 5–8 μm, polyblastic, discrete, terminal, subglobose to ellipsoidal, light brown, located at nodose apices of conidiophores.	13–15 × 12.5–14.5 μm, solitary or catenate, globose to subglobose, orangish brown to brown, aseptate, echinulate or verruculose	This study
BILAS 50335/ culture S2-11P (Sporulated on MEA)	Variable in length, simple, micro- or semi-macronematous, unbranched and branched, initially subhyaline to brownish, becoming dark brown at maturity, subhyaline or hyaline at the apex, septate, verruculose, formed singly or in small groups pushing through the weft of mycelium.	Mono- and polyblastic, discrete, determinate, terminal or lateral, subglobose, smooth to verruculose, pale brown, formed on an apical cell and in the collar region around the septa.	(11.5–)12.5–15(−17) μm diam., globose, pale brown to brown, verruculose or verrucose, in acropetal chains (3–4 in number).	[24]
MFLUCC 17-2292	350–420 × 4.5–5.5 μm, macronematous, mononematous, single or rarely 2–3 together on stroma, brown to dark brown, erect, or bent, septate, smooth, thick-walled.	Monoblastic, discrete on stipe.	9–12 × 8–12 μm, catenate, globose, brown to dark brown, aseptate, verruculose.	[21]
MFLUCC 19-0134	300–370 × 4–5 μm, macronematous, mononematous, unbranched, erect, single, light brown to dark brown, septate, smooth to minutely verruculose, thick-walled.	Monoblastic, proliferating, hyaline, terminal, blunt end, ovoid to globose, thick-walled.	11–13 × 10–12 μm, solitary, subglobose to globose, light brown to dark brown, aseptate, finely verruculose.	[40]
MFLUCC 20-0172 (as *P. celtidis*)	300–380 × 3.5–4.8 μm, macronematous, mononematous, unbranched, erect, single, greyish brown to dark brown, septate, smooth to minutely verruculose, thick-walled.	Monoblastic, proliferating, hyaline, terminal, blunt end, ovoid to globose, thick-walled.	8–10 × 9–10 μm diam., solitary, subglobose to globose, light brown to dark brown, aseptate, verruculose.	[40]

**Table 5 jof-08-00243-t005:** Polymorphic nucleotides from the ITS, LSU, and *tef1-α* sequence data of *Periconia pseudobyssoides*. The newly generated strain is indicated in bold black and the type strain is indicated as superscript “T”.

Species Name	Strain Number	Nucleotide Base Differences													
		ITS (Base Position)			LSU (Base Position)	*tef1-α* (Base Position)									
		76	419	522	371	331	405	471	564	570	579	648	720	725	764
*Periconia pseudobyssoides* ^T^	BILAS 50334	-	T	G	Sequence unavailable	Sequence unavailable									
*Periconia pseudobyssoides*	DUCC 0850	-	T	G	C	G	T	C	T	T	C	C	T	G	C
*Periconia pseudobyssoides*	KUMCC 20-0263	A	C	A	T	A	C	C	C	T	T	T	T	A	C
*Periconia pseudobyssoides*	MAFF 243868	-	T	G	C	A	T	T	C	T	C	T	C	A	G
*Periconia pseudobyssoides*	MAFF 243874	-	T	G	C	G	T	C	C	C	C	C	T	A	C

## Data Availability

Not applicable.

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
