# Peer review of "Taxonomic Reappraisal of Periconiaceae with the Description of Three New Periconia Species from China"

_jof, 2022, doi:10.3390/jof8030243_

Round 1

Reviewer 1 Report

Journal of Fungi 

Review 

The Manuscript (ID: jof-1594292), entitled " Taxonomic reappraisal of Periconia (Periconiaceae, Massarineae,

Pleosporales)”, authors: Er-Fu Yang, Rungtiwa Phookamsak, Hong-Bo Jiang, Saowaluck Tibpromma, Darbhe J. Bhat, Samantha C. Karunarathna, Dong-Qin Dai, Jianchu Xu, Itthayakorn Promputtha , Submitted to section: Environmental and Ecological Interactions of Fungi, meets the standard criteria required by Journal of Fungi. The manuscript is interesting taxonomic study, provides descriptions of three new Periconia species found in China (P. artemisiae, P. chimonanthi and P. thysanolaenae) and new, very important data on the hosts range and distribution of some previously described Periconia species - P. byssoides and P. pseudobyssoides. The results of this study are confirmed by molecular data and new taxonomical proposals are based on detailed morphological comparison of all known Periconia, Bambusistroma and Noosia species and phylogenetic analyses of a combined LSU, ITS, SSU and tef1-α sequence dataset. In this study authors also proposed new taxonomical recombination between P. byssoides and P. celtidis, synonymized Bambusistroma and Noosia under Periconia. The descriptions of new species are very informative and detailed, well-illustrated. The phylogenetic tree of Periconiaceae using maximum likelihood and Bayesian inference analyses is also provided. I recommend this manuscript for publication as an original paper. But I suggest correcting the title of this paper because it is not very suitable. In this study authors analysed mainly pure cultures of Periconia collected in China. I suggest changing the title, for example, to something such as „ Some taxonomic reappraisal of Periconiaceae with the description of three new Periconia species from China “.

My proposal - Accept after Minor Revisions.

However, I have few comments and questions, listed below.

Firstly, I suggest to correct the title.               

Abstract

Line 29. Please don‘t use the term „hyphomycetes“ and change it to „anamorphic ascomyces“.

Line 38. I propose to separate this sentense „........P.byssoides in the present phylogenetic analyses. while Results of .....“

Line 44. Correct „...sporulated the brown conidial hyphomycetes“ to „..produced brown conidia of asexual morph on agar media“

Line 46-47. Please correct this sentense. You have not provided detailed descriptions and illustrations to all known species of Periconiaceae.

Keywords: I propose reject „hyphomycetes“ and insert –„Ascomycota“. Also correct „Chinese mycota“ to „Chinese mycobiota“

                             Introduction

Line 51. Please clarify taxonomic position of Periconiaceae among Ascomycota.

Line 101-102. Please clarify the number of species and insert citation (Line 103) after „... on molecular data (?...)“ In the end of this manuscript (Line 844) the number is different.

Line 115. „Currently, only a putative species is accomodated in this genus“. Please clarify which one species?

Line 117. Bambusistroma – must be in Italic.

Line 118. Please correct „..the asexual hyphomycetous genus..“ to „asexual genus Noosia“

Results

Line 282. Clarify the name P. verrocosa?

Line 380, 492, 563, I propose to reject the term „hyphomycetous“

Line 481, Table 2. Why you do not included the sinopsis of Periconia thysanolaenae sp. nov. into this table? In my oppinion, that it is not corret to give sinopsis only in vitro (on PDA or on other media) because these species produce conidiophores on hosts. If you want to give Table with sinopsis of Periconia species found in China, it will be better to give full sinopsis of asexual and sexual morph of all described in this paper species, indicating characteristics on hosts and on media.

Line 547. Please correct the title of Table 4. May be -  Sinopsis of P. byssoids from different habitats? It is not clear what is the meaning of  –„  ....  (based on molecular data).

Line 645. ..“...Sporidesmium tengii was introduced by ?...(69) and is characterised..“ Please clarify the author.

Line 375, 438, 557, 620, 729. I propose the names of new Periconia species and new combinations indicate in Bold

Line 665 Please separate „..on dead stalks of Heracleum sosnowskyi...“

Line 705-706. Please clarify which hosts were from terrestrial habitats.

Line 768. Please clarify the habitat. In terrestrial habitat?

Line 844. Please clarify the data. „ Out of 119 morphologically.......“ In the Introduction (Line 101-102) you mentioned 117 morphological species.

Please clarify the taxon name Morosphaeria velatispora in the Table 1.

Please correct in the Table 1. the name „Periconia celtidis“ to „Periconia byssoides (as Periconia celtidis), similarly to Periconia banksiae (as Noosia banksiana), because in the phylogenetic tree I see the name Periconia byssoides „Periconia celtidis“

Figure 1 . Please clarify the taxon name ?Moroxhpaeria velatispora

Figure 2. It is not clear which strains were previously identified as Periconia celtidis.

Author Response

Dear Reviewer,

Thank you very much for your valuable comments and suggestions.

We have done editing our manuscript following the reviewer point by point.

Please see the revised manuscript in the attached file and the revisions have proceeded with track changes.

If you need us to include more information, please feel free to let us know.

Best regards,

Itthayakorn Promputtha

Reviewer 2 Report

In my opinion, it is a very nice "piece of paper". Very nice illustrated and phylogenetic analysis were well conducted and described. I have included some suggestions directly in the PDF file and the Authors will be able to see each of them. Keep studying the taxonomy of very beautiful, interesting and 'complicated' fungi as Periconia-like.

Author Response

(The authors gave the same response as above.)
